# EAGLE 🦅: Efficient Adaptive Geometry-based Learning in Cross-view Understanding

**Thanh-Dat Truong[1], Utsav Prabhu[2], Dongyi Wang[3]**
**Bhiksha Raj[4,5], Susan Gauch[6], Jeyamkondan Subbiah[7], Khoa Luu[1]**
[1]CVIU Lab, University of Arkansas, USA    [2]Google DeepMind, USA
[3]Dep. of BAEG, University of Arkansas, USA    [4]Carnegie Mellon University, USA
[5]Mohammed bin Zayed University of AI, UAE
[6]Dep. of EECS, University of Arkansas, USA    [7]Dep. of FDSC, University of Arkansas, USA
`{tt032, dongyiw, sgauch, jsubbiah, khoaluu}@uark.edu`
`bhiksha@cs.cmu.edu, utsavprabhu@google.com`
https://uark-cviu.github.io/projects/EAGLE

## Abstract

Unsupervised Domain Adaptation has been an efficient approach to transferring the semantic segmentation model across data distributions. Meanwhile, the recent Open-vocabulary Semantic Scene understanding based on large-scale vision language models is effective in open-set settings because it can learn diverse concepts and categories. However, these prior methods fail to generalize across different camera views due to the lack of cross-view geometric modeling. At present, there are limited studies analyzing cross-view learning. To address this problem, we introduce a novel Unsupervised Cross-view Adaptation Learning approach to modeling the geometric structural change across views in Semantic Scene Understanding. First, we introduce a novel Cross-view Geometric Constraint on Unpaired Data to model structural changes in images and segmentation masks across cameras. Second, we present a new Geodesic Flow-based Correlation Metric to efficiently measure the geometric structural changes across camera views. Third, we introduce a novel view-condition prompting mechanism to enhance the view-information modeling of the open-vocabulary segmentation network in cross-view adaptation learning. The experiments on different cross-view adaptation benchmarks have shown the effectiveness of our approach in cross-view modeling, demonstrating that we achieve State-of-the-Art (SOTA) performance compared to prior unsupervised domain adaptation and open-vocabulary semantic segmentation methods.

## 1 Introduction

Modern segmentation models [3, 4, 63] have achieved remarkable results on the close-set training with a set of pre-defined categories and concepts. To work towards human-level perception where the scenes are interpreted with diverse categories and concepts, the open-vocabulary (open-vocab) perception model [38, 40] based on the power of large vision-language models [30, 39] has been introduced to address the limitations of close-set training. By using the power of language as supervision, the large-scale vision language model is able to learn the more powerful representations where languages offer better reasoning mechanisms and open-word concept representations compared to traditional close-set training methods [3, 63, 9].

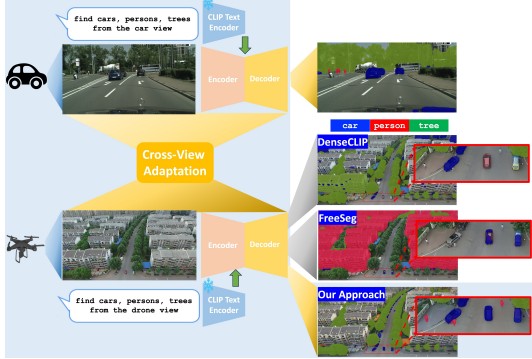

Figure 1: *Our Proposed Cross-view Adaptation Learning Approach.* Prior models, e.g., FreeSeg [38], DenseCLIP [40], trained on the car view do not perform well on the drone-view images. Meanwhile, our cross-view adaptation approach is able to generalize well from the car to drone view.

38th Conference on Neural Information Processing Systems (NeurIPS 2024).

Recent work is inspired by the success of large vision-language models [39, 27] that are able to learn informative feature representations of both visual and textual inputs from large-scale image-text pairs. These have been adopted to further develop open-vocab semantic segmentation models [38, 40, 31, 29] that can work well in open-world environments. However, the open-vocab perception models remain unable to generalize across camera viewpoints. As shown in Fig. 1, the open-vocab model trained on car views is not able to perform well on the images captured from unmanned aerial vehicles (UAVs) or drones. While this issue can be improved by training the segmentation model on drone-view

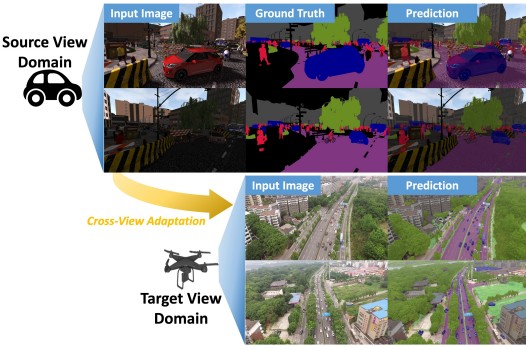

Figure 2: An Example of Illustration of Cross-View Adaptation From Car View to Drone View.

data, the annotation process of high-resolution UAV data is costly and time-consuming. At present, there exist many large-scale datasets with dense labels captured from camera views on the ground, e.g., car views (SYNTHIA [44], GTA [43], Cityscapes [11], BDD100K [68]). They have been widely adopted to develop robust perception models. Since these car view and drone view datasets have many common objects of interest, incorporating knowledge from car views with drone views benefits the learning process by reusing large-scale annotations and saving efforts of manually labeling UAV images. Unsupervised domain adaptation (UDA) [58, 23, 1, 51, 53] is one of the potential approaches to transfer the knowledge from the car view (i.e., source domain) to the drone view (i.e., target domain). While UDA approaches have shown their effectiveness in transferring knowledge across domains, e.g., environment changes or geographical domain shifts, these methods remain limited in the cases of changing camera viewpoints. Indeed, the changes in camera positions, e.g., from the ground of cars to the high positions of drones, bring a significant difference in structures and topological layouts of scenes and objects (Fig. 2). Therefore, UDA is not a complete solution to this problem due to its lack of cross-view structural modeling. Additionally, although the open-vocab segmentation models have introduced several prompting mechanisms, e.g., context-aware prompting [40] or adaptive prompting [38] to improve context learning across various open-world concepts, they are unable to model the cross-view structure due to the lack of view-condition information in prompts and geometric modeling. To the best of our knowledge, there are limited studies that have exploited this cross-view learning. These limitations motivate us to develop a new adaptation learning paradigm, i.e., *Unsupervised Cross-view Adaptation*, that addresses prior methods to improve the performance of semantic segmentation models across views.

**Contributions:** This work introduces a novel *Efficient Adaptive Geometry-based Learning (EAGLE)* to *Unsupervised Cross-view Adaptation* that can adaptively learn and improve the performance of semantic segmentation models across camera viewpoints. First, by analyzing the geometric correlations across views, we introduce a novel *cross-view geometric constraint* on *unpaired data* of structural changes in images and segmentation masks. Second, to efficiently model *cross-view geometric structural changes*, we introduce a new *Geodesic Flow-based Metric* to measure the structural changes across views via their manifold structures. In addition, to further improve the prompting mechanism of the open-vocab segmentation network in cross-view adaptation learning, we introduce a new *view-condition prompting*. Then, our *cross-view geometric constraint* is also imposed on its feature representations of view-condition prompts to leverage its geometric knowledge embedded in our prompting mechanism. Our proposed method holds a promise to be an effective approach to addressing the problem of cross-view learning and contributes to improving UDA and open-vocab segmentation in cross-view learning. Thus, it increases the generalizability of the segmentation models across camera views. Finally, our experiments on three presented cross-view adaptation benchmarks, i.e., SYNTHIA → UAVID, GTA → UAVID, BDD → UAVID, illustrate the effectiveness of our approach in cross-view modeling and our State-of-the-Art (SOTA) performance.

## 2 Related Work

**Unsupervised Domain Adaptation** Adversarial learning [6, 58, 59] and self-supervised learning [1, 69, 23, 14] are common approaches to UDA in semantic segmentation. The adversarial learning approaches are typically simultaneously trained on source and target data [58, 7, 6]. Chen et al. [7] first introduced an adversarial framework to domain adaptation. Later, several approaches improved

adversarial learning by utilizing generative models [74, 34, 21], using additional labels [28, 59], incorporating with entropy minimization [58, 65, 51, 52], or adopting the curriculum training [37]. Recently, the self-supervised approaches [1, 69, 23, 14] have achieved outstanding performance. Araslanov et al. [1] first proposed a self-supervised augmentation consistency framework for UDA. Hoyer et al. [23] utilized Transformers to improve the UDA performance. Later, this approach was further improved by utilizing multi-resolution cropped images [24] and masked image consistency strategy [25] to enhance contextual learning. Recent studies improved the self-supervised approach by aligning both output and attention levels via the cross-domain prediction consistency framework [60], using a prototypical representation [69], learning the cross-model consistency via depths [67], improving the class-relevant fairness [53, 55, 56], or exploring the relations of pseudo-labels [71]. Fashes et al. [15] introduced a prompt-based feature augmentation method to zero-shot UDA. Gong et al. [18] introduced a geodesic flow kernel to model the manifold structure between domains. Later, Simon et al. [47] designed distillation loss by the geodesic flow path.

**Vision-Language and Open-Vocab Segmentation** By pre-training on a large-scale vision-language dataset [39, 27], the vision-language models can learn various visual concepts and can further be transferred to other vision problems through "***prompting***" [17, 38, 31, 35], e.g., open-vocab segmentation [64, 13, 38]. Li et al. [29] first introduced the language-driving approach to semantic segmentation. Rao et al. [40] represented a context-aware prompting mechanism for dense prediction tasks. Ghiasi et al. [17] proposed an OpenSeg framework that learns the visual-semantic alignments. Qin et al. [38] presented a unified, universal, and open-vocab segmentation network based on Mask2Former [8] with an adaptive prompting mechanism. Xu et al. [64] proposed a two-stage open-vocab segmentation framework using the mask proposal generator and the pre-trained CLIP model. Ding et al. [13] decoupled the zero-shot semantic segmentation to class-agnostic segmentation and segment-level zero-shot classification. Liang et al. [31] improved the two-stage open-vocab segmentation model by further fine-tuning CLIP on masked image regions and corresponding descriptions.

**Cross-view Learning** The early studies exploited cross-view learning in geo-localization by using a polar transform across views [46, 45] or generative networks to cross-view images [41, 49]. Meanwhile, Zhu et al. [75] exploited the correlation between street- and aerial-view data via self-attention. In semantic segmentation, Coors et al. [10] first introduced a cross-view adaptation approach utilizing the depth labels and the cross-view transformation between car and truck views. However, this change of views in [10] is not as big a hurdle as the change of views in our problem, i.e., car view to drone view. Ren et al. [42] presented an adaptation approach across viewpoints using the 3D models of scenes to create pairs of cross-view images. Vidit et al. [57] modeled the geometric shift in cross FoV setting for object detection by learning position-invariant homography transform. Di Mauro et al. [12] introduced an adversarial method trained on a multi-view synthetic dataset where images are captured from different pitch and yaw angles at the same altitudes of the camera positions. Meanwhile, in our problem, the camera views could be placed at different altitudes (e.g., the car and the drone), which reveals large structural differences between the images. Truong et al. [50, 54] first introduced a simple approach to model the relation across views. CROVIA [50] measures the cross-view structural changes by measuring the distribution shift and only focuses on the cross-view adaptation setting in semantic segmentation. However, these methods [50, 54] lack a theory and a mechanism for cross-view geometric structural change modeling. To the best of our knowledge, there are limited studies exploiting cross-view adaptation in semantic segmentation. Therefore, our work presents a new approach to model the geometric correlation across views.

## 3 The Proposed EAGLE Approach

In this paper, we consider cross-view adaptation learning as UDA where the images of the source and target domains are captured from different camera positions (Fig. 2). Formally, let $\mathbf{x}_s, \mathbf{x}_t$ be the input images in the source and target domains, $\mathbf{p}_s, \mathbf{p}_t$ be the the corresponding prompts, and $\mathbf{y}_s, \mathbf{y}_t$ be the segmentation masks of $\mathbf{x}_s, \mathbf{x}_t$. Then, the open-vocab segmentation model $F$ maps the input $\mathbf{x}$ and the prompt $\mathbf{p}$ to the corresponding output $\mathbf{y} = F(\mathbf{x}, \mathbf{p})$. It should be noted that in the case of traditional semantic segmentation, the prompt $\mathbf{p}$ will be ignored, i.e., $\mathbf{y} = F(\mathbf{x})$ The cross-view adaptation learning can be formulated as Eqn. (1).

$$\arg \min_{\theta} \left[ \mathbb{E}_{\mathbf{x}_s, \mathbf{p}_s, \hat{\mathbf{y}}_s} \mathcal{L}_{Mask}(\mathbf{y}_s, \hat{\mathbf{y}}_s) + \mathbb{E}_{\mathbf{x}_t, \mathbf{p}_t} \mathcal{L}_{Adapt}(\mathbf{y}_t) \right] \tag{1}$$

where $\theta$ is the parameters of $F$, $\hat{\mathbf{y}}_s$ is the ground truth, $\mathcal{L}_{Mask}$ is the supervised (open-vocab) segmentation loss with ground truths, and $\mathcal{L}_{Adapt}$ is unsupervised adaptation loss from the source to the target domain. In the open-vocab setting, we adopt the design of Open-Vocab Mask2Former [8, 38] to our network $F$. Prior UDA methods defined the adaptation loss $\mathcal{L}_{Adapt}$ via the adversarial loss [28, 5], entropy loss [51, 58], or self-supervised loss [23, 25]. Although these prior results have illustrated their effectiveness in UDA, these losses remain limited in cross-view adaptation setup. Indeed, the adaptation setting in prior studies [58, 1, 23, 15] is typically deployed in the context of environmental changes (e.g., simulation to real [58, 59, 15], day to night [25, 15], etc) where the camera positions between domains remain similar. Meanwhile, in cross-view adaptation, the camera position of the source and target domain remains largely different (as shown in Fig. 2). This change in camera positions leads to significant differences in the geometric layout and topological structures between the source and target domains. As a result, direct adoption of prior UDA approaches to cross-view adaptation would be ineffective due to the lack of cross-view geometric correlation modeling. To effectively address cross-view adaptation, the adaptation loss $\mathcal{L}_{Adapt}$ should be able to model (1) *the geometric correlation between two views of source and target domains* and (2) *the structural changes across domains*.

## 3.1 Cross-View Geometric Modeling

To efficiently address the cross-view adaptation learning task, it is essential to explicitly model cross-view geometric correlations by analyzing the relation between two camera views. Therefore, we first re-reconsider the cross-view geometric correlation. In particular, let $\bar{\mathbf{x}}_t$ be the corresponding image of $\mathbf{x}_s$ captured from the target view, $\mathbf{y}_s$ and $\bar{\mathbf{y}}_t$ be the semantic segmentation outputs of source image $\mathbf{x}_s$ and target image $\bar{\mathbf{x}}_t$, $\bar{\mathbf{p}}_t$ be the corresponding prompt of $\mathbf{p}_s$ in target view, respectively. Formally, the images captured from the source and the target views can be modeled as Eqn. (2).

$$\mathbf{x}_s = \mathcal{R}(\mathbf{K}_s, [\mathbf{R}_s, \mathbf{t}_s], \Theta), \quad \bar{\mathbf{x}}_t = \mathcal{R}(\mathbf{K}_t, [\mathbf{R}_t, \mathbf{t}_t], \Theta) \tag{2}$$

where $\mathcal{R}$ is the rendering function, $\mathbf{K}_s$ and $\mathbf{K}_t$ are the intrinsic matrices, $[\mathbf{R}_s, \mathbf{t}_s]$ and $[\mathbf{R}_t, \mathbf{t}_t]$ are the extrinsic matrices, and $\Theta$ represents the capturing scene. In addition, as the camera parameters of both source and target views are represented by matrices, there should exist linear transformations of camera parameters between two views as follow,

$$\mathbf{K}_t = \mathbf{T_K} \times \mathbf{K}_s, \quad [\mathbf{R}_t, \mathbf{t}_t] = \mathbf{T_{Rt}} \times [\mathbf{R}_s, \mathbf{t}_s] \tag{3}$$

where $\mathbf{T_K}$ and $\mathbf{T_{Rt}}$ are the transformation matrices.

***Remark 1: The Geometric Transformation Between Camera Views.*** From Eqn. (2) and Eqn. (3), we argue that there should exist a geometric transformation $\mathcal{T}$ of images between two camera views as: $\bar{\mathbf{x}}_t = \mathcal{T}(\mathbf{x}_s; \mathbf{T_K}, \mathbf{T_{Rt}})$.

***Remark 2: The Equivalent Transformation Between Image and Segmentation Output.*** As RGB images and segmentation maps are pixel-wised corresponding, the same geometric transformation $\mathcal{T}$ in the image space can be adopted for segmentation space as: $\bar{\mathbf{y}}_t = \mathcal{T}(\mathbf{y}_s; \mathbf{T_K}, \mathbf{T_{Rt}})$

Remarks 1-2 have depicted that the geometric transformation of both image and segmentation from the source to the target view can be represented by the shared transformation $\mathcal{T}$ with the camera transformation matrices $\mathbf{T_K}, \mathbf{T_{Rt}}$. Let $\mathcal{D}_x(\mathbf{x}_s, \bar{\mathbf{x}}_t)$ and $\mathcal{D}_y(\mathbf{y}_s, \bar{\mathbf{y}}_t)$ *be the metrics the measure the cross-view structures changes* of images and segmentation maps from the source to target domains.

We argue that the cross-view geometric correlation in the image space, i.e., $\mathcal{D}_x(\mathbf{x}_s, \bar{\mathbf{x}}_t)$, is theoretically proportional to the one in the segmentation space, i.e., $\mathcal{D}_y(\mathbf{y}_s, \bar{\mathbf{y}}_t)$. Since the camera transformations between the two views are linear (Eqn. (3)) and the images $\mathbf{x}$ and outputs $\mathbf{y}$ are pixel-wised corresponding, we hypothesize that the cross-view geometric correlation in the image space $\mathcal{D}_x(\mathbf{x}_s, \bar{\mathbf{x}}_t)$ and the segmentation space $\mathcal{D}_y(\mathbf{y}_s, \bar{\mathbf{y}}_t)$ can be modeled by a linear relation with linear scale $\alpha$ as follows:

$$\mathcal{D}_x(\mathbf{x}_s, \bar{\mathbf{x}}_t) \propto \mathcal{D}_y(\mathbf{y}_s, \bar{\mathbf{y}}_t) \Leftrightarrow \mathcal{D}_x(\mathbf{x}_s, \bar{\mathbf{x}}_t) = \alpha \mathcal{D}_y(\mathbf{y}_s, \bar{\mathbf{y}}_t) \tag{4}$$

## 3.2 Cross-view Geometric Learning on Unpaired Data

Eqn. (4) defines a necessary condition to explicitly model the cross-view geometric correlation. Therefore, cross-view adaptation learning in Eqn. (1) can be re-formed as follows:

$$\theta^* = \arg\min_{\theta} \left[ \mathbb{E}_{\mathbf{x}_s, \mathbf{p}_s, \hat{\mathbf{y}}_s} \mathcal{L}_{Mask}(\mathbf{y}_s, \mathbf{p}_s, \hat{\mathbf{y}}_s) + \mathbb{E}_{\mathbf{x}_s, \mathbf{p}_s, \bar{\mathbf{x}}_t, \bar{\mathbf{p}}_t} || \mathcal{D}_x(\mathbf{x}_s, \bar{\mathbf{x}}_t) - \alpha \mathcal{D}_y(\mathbf{y}_s, \bar{\mathbf{y}}_t) || \right] \tag{5}$$

where, $\mathcal{L}_{Adapt}(\mathbf{y}_s, \bar{\mathbf{y}}_t) = ||\mathcal{D}_x(\mathbf{x}_s, \bar{\mathbf{x}}_t) - \alpha\mathcal{D}_y(\mathbf{y}_s, \bar{\mathbf{y}}_t)||$ is the cross-view geoemtric adaptation loss, $||\cdot||$ is the mean squared error loss. However, in practice, the pair data between source and target views are inaccessible as data from these two views are often collected independently. Thus, optimizing Eqn. (5) without cross-view pairs of data remains an ill-posed problem. To address this limitation, instead of learning Eqn. (5) on paired data, we proposed to model this correlation on unpaired data. Instead of solving the cross-view geometric constraint of Eqn. (5) on pair data, let us consider all cross-view unpaired samples $(\mathbf{x}_s, \mathbf{x}_t)$. Formally, learning the ***Cross-view Geometric Constraint*** between unpaired samples can be formulated as in Eqn. (6).

$$\theta^* = \arg\min_\theta \left[ \mathbb{E}_{\mathbf{x}_s, \hat{\mathbf{y}}_s} \mathcal{L}_{Mask}(\mathbf{y}_s, \mathbf{p}_s, \hat{\mathbf{y}}_s) + \mathbb{E}_{\mathbf{x}_s, \mathbf{p}_s, \mathbf{x}_t, \mathbf{p}_t} ||\mathcal{D}_x(\mathbf{x}_s, \mathbf{x}_t) - \alpha\mathcal{D}_y(\mathbf{y}_s, \mathbf{y}_t)|| \right] \tag{6}$$

where $\mathbf{x}_s$ and $\mathbf{x}_t$ are unpaired data, and $\mathcal{L}_{Adapt}(\mathbf{y}_s, \mathbf{y}_t) = ||\mathcal{D}_x(\mathbf{x}_s, \mathbf{x}_t) - \alpha\mathcal{D}_y(\mathbf{y}_s, \mathbf{y}_t)||$ is the ***Cross-view Geometric Adaptation*** loss on unpaired data. Intuitively, although the cross-view pair samples are not available, the cross-view geometric constraints on paired samples between two views can be indirectly imposed by modeling the cross-view geometric structural constraint among unpaired samples. Then, by modeling the cross-view structural changes in the image and segmentation spaces, the structural change on images of unpaired data could be considered as the reference for the cross-view structural change in the segmentation space during the optimization process. This action promotes the structures of segmentation that can be effectively adapted from the source view to the target view. Importantly, the cross-view geometric constraint imposed on unpaired data can be mathematically proved as an upper bound of the cross-view constraint on paired data as follows:

$$||\mathcal{D}_x(\mathbf{x}_s, \bar{\mathbf{x}}_t) - \alpha\mathcal{D}_y(\mathbf{y}_s, \bar{\mathbf{y}}_t)|| = \mathcal{O}\left(\mathcal{D}_x(||\mathbf{x}_s, \mathbf{x}_t) - \alpha\mathcal{D}_y(\mathbf{y}_s, \mathbf{y}_t)||\right) \tag{7}$$

where $\mathcal{O}$ is the Big O notation. The upper bound in Eqn. (19) can be proved by using the properties of triangle inequality and our correlation metrics $\mathcal{D}_x$ and $\mathcal{D}_y$ (Sec. 3.3). The detailed proof is provided in the appendix. Eqn. (19) has illustrated that by minimizing the cross-view geometric constraint on unpaired samples in Eqn. (6), the cross-view constraint on paired samples in Eqn. (5) is also maintained due to the upper bound. Therefore, our proposed Cross-view Geometric Constraint loss ***does NOT require the pair data between source and target views*** during training. Fig. 3 illustrates our cross-view adaptation learning framework.

### 3.3 Cross-view Structural Change Modeling via Geodesic Flow Path

Modeling the correlation metrics $\mathcal{D}_x$ and $\mathcal{D}_y$ is an important task in our approach. Indeed, the metrics should be able to model the structure changes from the source to the target view. Intuitively, the changes from the source to the target view are essentially the geodesic flow between two subspaces on the Grassmann manifold. Then, the images (or segmentation) of two views can be projected along the geodesic flow path to capture the cross-view structural changes. Therefore, to model $\mathcal{D}_x$ and $\mathcal{D}_y$, we adopt the ***Geodesic Flow*** path to measure the cross-view structural changes by modeling the geometry in the latent space.

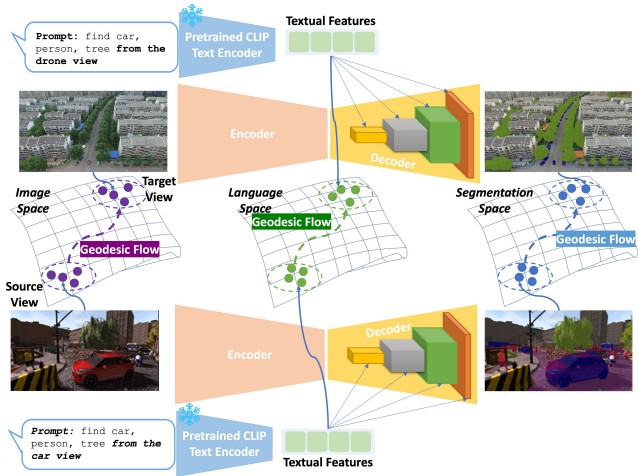

Figure 3: **Our Cross-View Learning Framework.**

**Remark 3: Grassmann Manifold** is the set of $N$-dimensional linear subspaces of $\mathbb{R}^D (0 < N < D)$, i.e, $\mathcal{G}(N, D)$. A matrix with orthonormal columns $\mathbf{P} \in \mathbb{R}^{D \times N}$ define a subspace of $\mathcal{G}(N, D)$, i.e., $\mathbf{P} \in \mathcal{G}(N, D) \Rightarrow \mathbf{P}^\top\mathbf{P} = \mathbf{I}_N$ where $\mathbf{I}_N$ is the $N \times N$ identity matrix.

For simplicity, we present our approach to model the cross-view structural change $\mathcal{D}_x$ in the image space. Formally, let $\mathbf{P}_s$ and $\mathbf{P}_t$ be the basis of the source and target domains. These bases can be obtained by the PCA algorithm. The geodesic flow between $\mathbf{P}_s$ and $\mathbf{P}_t$ in the manifold can be defined via the function $\mathbf{\Pi} : \nu \in [0..1] \to \mathbf{\Pi}(\nu)$, where $\mathbf{\Pi}(\nu) \in \mathcal{G}(N, D)$ is the subspace lying on

the geodesic flow path from the source to the target view:

$$\mathbf{\Pi}(\nu) = [\mathbf{P}_s \quad \mathbf{R}][\mathbf{U}_1\mathbf{\Gamma}(\nu) \quad -\mathbf{U}_2\mathbf{\Sigma}(\nu)]^\top \tag{8}$$

where $\mathbf{R} \in \mathbb{R}^{D\times(D-N)}$ is the orthogonal complement of $\mathbf{P}_s$, i.e., $\mathbf{R}^\top\mathbf{P}_s = \mathbf{0}$. $\mathbf{\Gamma}(\nu)$ and $\mathbf{\Sigma}(\nu)$ are the diagonal matrices whose diagonal element at row $i$ can be defined as $\gamma_i = \cos(\nu\omega_i)$ and $\sigma_i = \sin(\nu\omega_i)$. The list of $\omega_i$ is the principal angles between source and target subspaces, i.e., $0 \le \omega_1 \le ... \le \omega_N \le \frac{\pi}{2}$. $\mathbf{U}_1$ and $\mathbf{U}_2$ are the orthonormal matrices obtained by the following pair of SVDs:

$$\mathbf{P}_s^\top\mathbf{P}_T = \mathbf{U}_1\mathbf{\Gamma}(1)\mathbf{V}^\top \qquad \mathbf{R}^\top\mathbf{P}_T = -\mathbf{U}_2\mathbf{\Sigma}(1)\mathbf{V}^\top \tag{9}$$

Since $\mathbf{P}_s^\top\mathbf{P}_t$ and $\mathbf{R}^\top\mathbf{P}_t$ share the same singular vectors $\mathbf{V}$, we adopt the generalized Singular Value Decomposition (SVD) [18, 47] to decompose the matrices. In our approach, we model the cross-view structural changes $\mathcal{D}_x$ by modeling the cosine similarity between projections along the geodesic flow $\mathbf{\Pi}(\nu)$. In particular, given a subspace $\mathbf{\Pi}(\nu)$ on the geodesic flow path from the source to the target view, the cross-view geometric correlation of images between the source and target views can formulated by the inner product $g_{\mathbf{\Pi}(\nu)}(\mathbf{x}_s, \mathbf{x}_t)$ along the geodesic flow $\mathbf{\Pi}(\nu)$ as follows:

$$g(\mathbf{x}_s, \mathbf{x}_t) = \int_0^1 g_{\mathbf{\Pi}(\nu)}(\mathbf{x}_s, \mathbf{x}_t)d\nu = \int_0^1 \mathbf{x}_s^\top\mathbf{\Pi}(\nu)\mathbf{\Pi}(\nu)^\top\mathbf{x}_t d\nu = \mathbf{x}_s^\top\left(\int_0^1 \mathbf{\Pi}(\nu)\mathbf{\Pi}(\nu)^\top d\nu\right)\mathbf{x}_t = \mathbf{x}_s^\top\mathbf{Q}\mathbf{x}_t \tag{10}$$

where $\mathbf{Q} = \int_0^1 \mathbf{\Pi}(\nu)\mathbf{\Pi}(\nu)^\top d\nu$. Intuitively, the matrix $\mathbf{Q}$ represents the manifold structure between the source to the target view. Then, Eqn. (10) measures the cross-view structural changes between the source and the target domain based on their manifold structures. The matrix $\mathbf{Q}$ can be obtained in a closed form [18, 47] as follows:

$$\mathbf{Q} = [\mathbf{P}_s\mathbf{U}_1 \quad \mathbf{R}\mathbf{U}_2]\begin{bmatrix}\mathbf{\Lambda}_1 & \mathbf{\Lambda}_2 \\ \mathbf{\Lambda}_2 & \mathbf{\Lambda}_3\end{bmatrix}\begin{bmatrix}\mathbf{U}_1^\top\mathbf{P}_s^\top \\ \mathbf{U}_2^\top\mathbf{R}^\top\end{bmatrix} \tag{11}$$

where $\mathbf{\Lambda}_1$, $\mathbf{\Lambda}_2$, and $\mathbf{\Lambda}_3$ are the diagonal matrices, whose diagonal elements at row $i$ can be defined as:

$$\lambda_{1,i} = 1 + \frac{\sin(2\omega_i)}{2\omega_i}, \ \lambda_{2,i} = \frac{\cos(2\omega_i) - 1}{2\omega_i}, \ \lambda_{3,i} = 1 - \frac{\sin(2\omega_i)}{2\omega_i} \tag{12}$$

In practice, we model the cross-view structural changes $\mathcal{D}_x$ via the cosine similarity along the geodesic flows. Finally, the cross-view structural changes $\mathcal{D}_x$ can be formulate as:

$$\mathcal{D}_x(\mathbf{x}_s, \mathbf{x}_t) = 1 - \frac{\mathbf{x}_s^\top\mathbf{Q}\mathbf{x}_t}{||\mathbf{Q}^{1/2}\mathbf{x}_s||||\mathbf{Q}^{1/2}\mathbf{x}_t||} \tag{13}$$

Similarly, we can model the cross-view geometric correlation of segmentation $\mathcal{D}_y$ via Geodesic Flow.

### 3.4 View-Condition Prompting to Cross-View Learning

**View-Condition Prompting** Previous efforts [40, 38, 16, 73] in open-vocab segmentation have shown that a better prompting mechanism can provide more meaningful textual and visual knowledge. Prior work in open-vocab segmentation designed the prompt via the class names [64, 13, 38], e.g., "$\texttt{class}_1, \texttt{class}_1, ..., \texttt{class}_K$". Meanwhile, other methods improve the prompting mechanism by introducing the learnable variables into the prompt [40] or adding the task information [38]. This action helps to improve the context learning of the vision-language model. In our approach, we also exploit the effectiveness of designing prompting to cross-view learning. In particular, describing the view information can further improve the visual context learning, e.g., "$\texttt{class}_1, \texttt{class}_1, ..., \texttt{class}_K$ captured from the [domain] view", where [domain] could be car (source domain) or drone (target domain). Therefore, we introduce a view-condition prompting mechanism by introducing the view information, i.e., captured from the [domain] view", into the prompt. Our view-condition prompt offers the context specific to visual learning, thus providing better transferability in cross-view segmentation.

**Cross-view Correlation of View-Condition Prompts** We hypothesize that the correlation of the input prompts across domains also provides the cross-view geometric correlation in their deep representations. In particular, let $\mathbf{f}_s^p$ and $\mathbf{f}_t^p$ be the deep textual embeddings of view-condition prompts $\mathbf{p}_s$ and $\mathbf{p}_t$, and $\mathcal{D}_p$ be metric measuring the correlation between $\mathbf{f}_s^p$ and $\mathbf{f}_t^p$. In addition, since the textual encoder has been pre-trained on large-scale vision-language data [39, 27], the visual and the textual representations have been well aligned. Then, we argue that the correlation of

Table 1: Effectiveness of Our Cross-view Adaptation Losses and Prompting Mechanism.

| With Prompt | Cross-View Adapt | View Condition | SYNTHIA → UAVID | | | | | | GTA → UAVID | | | | | | |
|---|---|---|---|---|---|---|---|---|---|---|---|---|---|---|---|
| | | | Road | Building | Car | Tree | Person | mIoU | Road | Building | Car | Tree | Terrain | Person | mIoU |
| ✗ | ✗ | ✗ | 8.1 | 19.1 | 7.4 | 30.3 | 1.3 | 13.2 | 7.5 | 13.0 | 2.7 | 26.8 | 26.6 | 1.0 | 12.9 |
| ✗ | ✓ | ✗ | **31.4** | **75.1** | **57.5** | **59.2** | **19.5** | **48.6** | **22.9** | **64.6** | **37.8** | **52.8** | **48.5** | **13.8** | **40.1** |
| | Supervised | | 75.5 | 91.6 | 79.1 | 77.7 | 42.1 | 73.2 | 76.8 | 91.8 | 81.1 | 77.6 | 67.8 | 43.4 | 73.1 |
| ✓ | ✗ | ✗ | 15.7 | 27.8 | 15.7 | 34.1 | 7.7 | 20.2 | 16.6 | 26.8 | 7.2 | 30.0 | 21.7 | 6.0 | 18.1 |
| ✓ | ✓ | ✗ | 36.8 | 75.5 | 61.3 | 60.8 | 21.2 | 51.1 | 27.3 | 66.8 | 42.3 | 55.5 | 47.1 | 25.1 | 44.0 |
| ✓ | ✓ | ✓ | **38.4** | **76.1** | **62.8** | **62.1** | **21.8** | **52.2** | **29.2** | **67.1** | **45.2** | **56.6** | **48.5** | **27.9** | **45.7** |
| | Supervised | | 79.8 | 92.6 | 82.9 | 79.1 | 48.0 | 76.5 | 80.5 | 93.3 | 82.7 | 79.2 | 71.3 | 49.9 | 76.1 |

textual feature representations across views, i.e., $\mathcal{D}_p(\mathbf{f}_s^p, \mathbf{f}_t^p)$, also provides the cross-view geometric correlation due to the embedded view information in the deep representation of prompts aligned with visual representations. Therefore, similar to Eqn. (4), we hypothesize the cross-view correlation of segmentation masks and textual features can be modeled as a linear relation with a scale factor $\gamma$ as:

$$\mathcal{D}_p(\mathbf{f}_s^p, \mathbf{f}_t^p) \propto \mathcal{D}_y(\mathbf{y}_s, \mathbf{y}_t) \Leftrightarrow \mathcal{D}_p(\mathbf{f}_s^p, \mathbf{f}_t^p) = \gamma \mathcal{D}_y(\mathbf{y}_s, \mathbf{y}_t) \tag{14}$$

Then, learning the cross-view adaptation with view-condition prompts can be formulated as follows:

$$\theta^* = \arg\min_\theta \left[ \mathbb{E}_{\mathbf{x}_s, \mathbf{p}_s, \hat{\mathbf{y}}_s} \mathcal{L}_{Mask}(\mathbf{y}_s, \hat{\mathbf{y}}_s) + \mathbb{E}_{\mathbf{x}_s, \mathbf{x}_s, \mathbf{x}_t, \mathbf{p}_t} \left( \lambda_I ||\mathcal{D}_x(\mathbf{x}_s, \mathbf{x}_t) - \alpha \mathcal{D}_y(\mathbf{y}_s, \mathbf{y}_t) + \lambda_P ||\mathcal{D}_p(\mathbf{f}_s^p, \mathbf{f}_t^p) - \gamma \mathcal{D}_y(\mathbf{y}_s, \mathbf{y}_t)||) \right] \tag{15}$$

where $\lambda_I$ and $\lambda_P$ are the balanced-weight of losses. Similar to metrics $\mathcal{D}_x$ and $\mathcal{D}_y$, we also adopt the geodesic flow path to model the cross-view correlation metric $\mathcal{D}_p$.

## 4 Experiments

### 4.1 Datasets, Benchmarks, and Implementation

To efficiently evaluate cross-view adaptation, the cross-view benchmarks are set up from the car to the drone view. Following common practices in UDA [23, 58], we choose SYNTHIA [44], GTA [43], and BDD100K [68] as the source domains while UAVID [33] is chosen as the target domain. We chose to adopt these datasets because they share a class of interests and are commonly used in UDA and segmentation benchmarks [23, 61].

**SYNTHIA → UAVID Benchmark** SYNTHIA and UAVID share five classes of interest, i.e., Road, Building, Car, Tree, and Person. Since the UAVID dataset annotated cars, trucks, and buses as a class of Car, we collapse these classes in SYNTHIA into a single class of Car.

**GTA → UAVID Benchmark** consists of five classes in the SYNTHIA → UAVID benchmark and includes one more class of Terrain. Therefore, the GTA → UAVID benchmark has six classes of interest, i.e., Road, Building, Car, Tree, Terrain, and Person.

**BDD → UAVID Benchmark** is a real-to-real cross-view adaptation setting. Similar to GTA → UAVID benchmark, there are six classes of interest between BDD100K and UAVID. In our experiments, we adopt the mean Intersection over Union (mIoU) metric to measure the performance.

**Implementation** We adopt Mask2Former [8] (ResNet 101) with Semantic Context Interaction of FreeSeg [38] and pre-trained text encoder of CLIP [39] for our open-vocab segmentation networks. Our balanced weights of losses are set to $\lambda_I = 1.0$ and $\lambda_P = 0.5$. Further details of our networks and hyper-parameters are provided in the appendix.

### 4.2 Ablation Study

**Effectiveness of Cross-view Adaptation and Prompting Mechanisms** Table 1 analyzes the effectiveness of prompting mechanisms, i.e., i.e., with and without ***Prompting***, with and without ***Cross-view Adaptation*** (in Eqn. (6)), with and without ***View-Condition Prompting*** (in Eqn. (15)). For supervised results, we train two different models on UAVID with and without the

Table 2: Effectiveness of Backbones and Cross-view Metrics.

| Network | Metric | SYNTHIA → UAVID | | | | | | |
|---|---|---|---|---|---|---|---|---|
| | | Road | Building | Car | Tree | Terrain | Person | mIoU |
| ResNet | Euclidean | 23.7 | 31.2 | 33.2 | 36.7 | - | 11.5 | 27.2 |
| | Geodesic | **38.4** | **76.1** | **62.8** | **62.1** | - | **21.8** | **52.2** |
| Swin | Euclidean | 24.7 | 31.9 | 41.2 | 39.7 | - | 14.1 | 30.3 |
| | Geodesic | **40.8** | **76.4** | **65.8** | **62.7** | - | **27.9** | **54.7** |
| | | GTA → UAVID | | | | | | |
| ResNet | Euclidean | 21.7 | 30.0 | 26.2 | 39.7 | 31.7 | 9.5 | 26.5 |
| | Geodesic | **29.2** | **67.1** | **45.2** | **56.6** | **48.5** | **27.9** | **45.7** |
| Swin | Euclidean | 24.3 | 33.7 | 28.5 | 40.1 | 32.8 | 9.7 | 28.2 |
| | Geodesic | **31.0** | **67.1** | **46.8** | **56.9** | **48.7** | **31.9** | **47.1** |

Terrain class on two benchmarks. As in Table 1, the cross-view adaptation loss in Eqn. (6) significantly improve the performance of segmentation models. With prompting and cross-view adaptation,

the mIoU performance is further boosted, i.e., the mIoU performance achieves $48.6\%$ and $40.1\%$ on two benchmarks. Additionally, by further using the view-condition prompting mechanism with our cross-view loss in Eqn. (15), the mIoU results are slightly improved by $+1.1\%$ and $+1.7\%$ on two benchmarks compared to the one without view-condition prompting. Our results have closed the gap with the upper-bound results where the models are trained on UAVID with labels.

**Effectiveness of Cross-view Correlation Metrics and Network Backbones** Table 2 studies the impact of choosing metrics and network backbones. We consider two options, i.e., Euclidean Metric and our Geodesic Flow-based Metric, for correlation metrics $\mathcal{D}_x$, $\mathcal{D}_y$, and $\mathcal{D}_p$. As shown in Table 2, our Geodesic Flow-based metrics significantly improve the performance of our cross-view adaptation. It has shown that our approach is able to measure the structural changes across views better than using the Euclidean metrics. In addition, by using the more powerful backbone (Swin), the performance of cross-view adaptation is further improved.

**Effectiveness of Cross-view Learning Parameters** Table 3 illustrates the impact of the linear scaling factors $\alpha$ and $\beta$. As in Table 3, the mIoU performance has been majorly affected by the relation between images and segmentation. The best performance is gained at the optimal value of $\alpha = 1.5$. Since the variation of RGB images is higher than the segmentation, the small value $\alpha$ could not correctly scale the rela-

Table 3: Effectiveness of Linear Scale Factors, i.e., $\alpha$ and $\gamma$, and Subspace dimension $D$.

| Factor | SYNTHIA → UAVID | | | | | | GTA → UAVID | | | | | | |
|---|---|---|---|---|---|---|---|---|---|---|---|---|---|
| | Road | Building | Car | Tree | Person | mIoU | Road | Building | Car | Tree | Terrain | Person | mIoU |
| $\alpha = 0.5$ | 35.9 | 73.6 | 59.6 | 57.2 | 20.8 | 49.4 | 25.4 | 63.9 | 40.5 | 44.6 | 46.8 | 25.6 | 41.1 |
| $\alpha = 1.0$ | 37.8 | 75.8 | 61.0 | 60.7 | 21.6 | 51.4 | 26.9 | 64.3 | 41.8 | 48.0 | 47.2 | 26.3 | 42.4 |
| $\alpha = 1.5$ | **38.4** | **76.1** | **62.8** | **62.1** | **21.8** | **52.2** | **29.2** | **67.1** | **45.2** | **56.6** | **48.5** | **27.9** | **45.7** |
| $\alpha = 2.0$ | 36.9 | 74.8 | 60.7 | 59.4 | 21.2 | 50.6 | 28.1 | 66.0 | 44.2 | 51.8 | 48.1 | 27.3 | 44.2 |
| $\gamma = 0.5$ | 37.6 | 75.5 | 60.6 | 60.0 | 21.4 | 51.1 | 27.8 | 65.1 | 42.7 | 51.7 | 47.7 | 26.8 | 43.6 |
| $\gamma = 1.0$ | **38.4** | **76.1** | **62.8** | **62.1** | **21.8** | **52.2** | **29.2** | **67.1** | **45.2** | **56.6** | **48.5** | **27.9** | **45.7** |
| $\gamma = 1.5$ | 36.2 | 75.3 | 61.6 | 58.5 | 20.5 | 50.4 | 28.5 | 66.0 | 44.2 | 54.7 | 47.9 | 27.5 | 44.8 |
| $\gamma = 2.0$ | 36.0 | 74.2 | 60.0 | 58.1 | 20.7 | 49.8 | 26.8 | 64.6 | 42.8 | 52.5 | 47.0 | 26.8 | 43.4 |
| $D = 96$ | 36.6 | 72.0 | 60.6 | 57.7 | 21.3 | 49.6 | 26.8 | 60.6 | 42.2 | 50.7 | 46.5 | 27.0 | 42.3 |
| $D = 128$ | 37.1 | 72.7 | 61.3 | 58.9 | 21.4 | 50.3 | 27.7 | 62.4 | 43.0 | 52.9 | 47.1 | 27.3 | 43.4 |
| $D = 256$ | **38.4** | **76.1** | **62.8** | **62.1** | **21.8** | **52.2** | **29.2** | **67.1** | **45.2** | **56.6** | **48.5** | **27.9** | **45.7** |
| $D = 512$ | 37.9 | 75.8 | 62.4 | 61.1 | 21.4 | 51.7 | 28.2 | 64.9 | 44.2 | 54.1 | 47.9 | 27.6 | 44.5 |

tion between images and segmentation while the higher value of $\alpha$ exaggerates the structural change of segmentation masks. Additionally, the change of $\gamma$ slightly affects the mIoU performance. Since the textual features are well-aligned with the image, the performance of segmentation models when changing $\gamma$ also behaves similarly to the changes of $\alpha$. However, the linear scale factor $\alpha$ is more sensitive to mIoU results since the images play a more important role in the segmentation results due to the pixel-wise corresponding of images and segmentation.

**Effectiveness of Subspace Dimension in Geodesic Flow** Table 3 reveals the importance of choosing the subspace dimension. The cross-view geometric structural change is better modeled by increasing the dimension of the subspaces. As in Table 3, the performance is improved when the dimension is increased from 96 to 256. However, beyond that point, the mIoU performance tends to be dropped. We have observed that low dimensionality cannot model the structural changes across views since it captures small variations in structural changes. Conversely, higher dimensionality includes more noise in the cross-view structural changes and increases the computational cost. We also study the impact of batch size in our appendix.

**Qualitative Results.** To further illustrate the effectiveness of our proposed, we visualize the results produced by our model. In the model without prompting, Figure 4 illustrates the results of our cross-view adaptation compared to those without adaptation. As shown in the results, our approach can effectively segment the objects in the drone view. We also compare with the prior ProDA [69] and CROVIA [50] methods. Our qualitative results remain better than the prior adaptation method. For the model with prompting, Figure 5 illustrates the effectiveness of our approach in three cases:

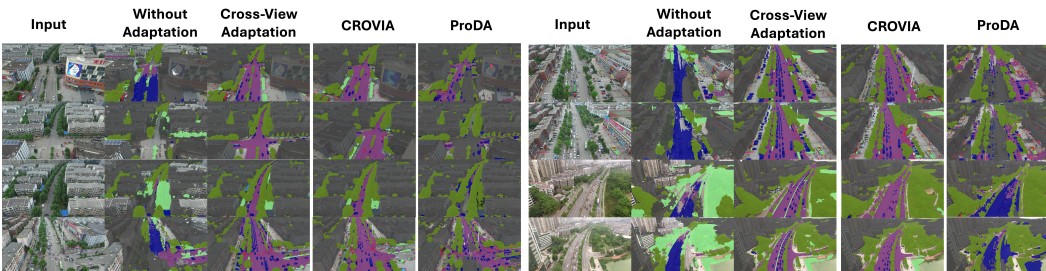

Figure 4: The Qualitative Results of Cross-View Adaptation (Without Prompt).

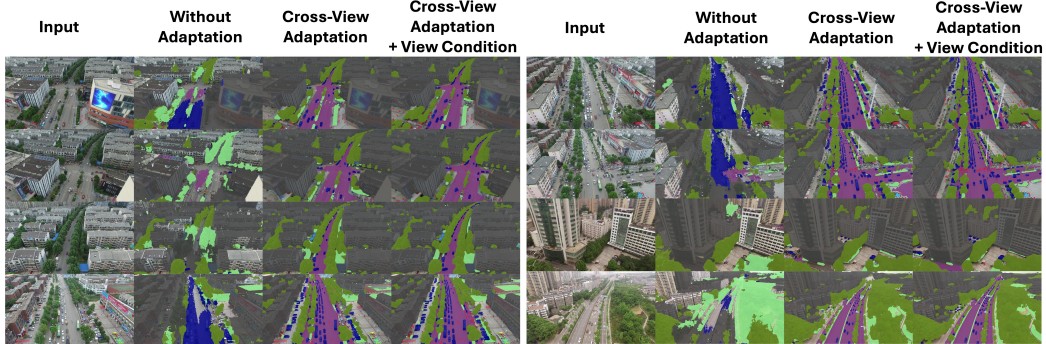

Figure 5: The Qualitative Results of Cross-View Adaptation (With Prompt).

without adaptation, with cross-view adaptation, and with view-condition prompting. As shown in the results, our cross-view adaptation can efficiently model the segmentation of the view. By using the view-condition prompting, our model can further improve the segmentation of persons and vehicles.

## 4.3 Comparisons with Prior UDA Methods

**SYNTHIA → UAVID** As shown in Table 4, our EAGLE has achieved SOTA results and outperforms prior view transformation (i.e., Polar Transform [45]) UDA methods by a large margin. For fair comparisons, we adopt the DeepLab [3] and DAFormer [23] for the segmentation network. In particular, our mIoU results using DeepLab and DAFormer are 45.2% and 50.8%. In the DAFormer backbone, the

Table 4: Comparisons with Domain Adaptation Approaches (Without Prompting).

| Network | Method | SYNTHIA → UAVID | | | | | | GTA → UAVID | | | | | | |
|---|---|---|---|---|---|---|---|---|---|---|---|---|---|---|
| | | Road | Building | Car | Tree | Person | mIoU | Road | Building | Car | Tree | Terrain | Person | mIoU |
| DeepLab | AdvEnt [58] | 4.7 | 63.2 | 31.7 | 48.6 | 11.4 | 31.9 | 2.0 | 30.3 | 14.9 | 29.8 | 41.5 | 1.8 | 20.0 |
| | Polar Trans. [45] | 20.5 | 10.9 | 38.2 | 22.6 | 4.3 | 19.3 | 19.4 | 9.1 | 37.8 | 20.7 | 15.6 | 2.5 | 17.5 |
| | DADA [59] | 10.7 | 63.1 | 32.9 | 50.0 | 16.2 | 34.6 | - | - | - | - | - | - | - |
| | BiMaL [51] | 5.4 | 62.1 | 34.8 | 50.7 | 12.7 | 33.1 | 1.3 | 44.6 | 10.1 | 49.2 | 20.0 | 10.9 | 22.7 |
| | SAC [1] | 13.9 | 64.0 | 18.7 | 48.0 | 15.6 | 32.0 | 4.5 | 36.9 | 7.8 | 47.9 | 44.1 | 7.8 | 24.8 |
| | ProDA [69] | 10.6 | 64.7 | 34.1 | 44.5 | 17.0 | 34.2 | 6.9 | 50.6 | 28.4 | 25.5 | 38.7 | 4.5 | 25.8 |
| | CROVIA [50] | 10.6 | 65.7 | 51.7 | 55.6 | 17.0 | 40.1 | 18.2 | 49.8 | 10.4 | 48.1 | 44.0 | 8.0 | 29.7 |
| | **EAGLE** | **29.9** | **65.7** | **55.5** | **56.8** | **18.3** | **45.2** | **20.5** | **53.0** | **37.6** | **50.7** | **45.3** | **13.0** | **36.7** |
| | Supervised | 67.2 | 90.7 | 74.0 | 76.3 | 36.8 | 69.0 | 68.1 | 91.0 | 77.5 | 75.7 | 62.2 | 35.8 | 68.4 |
| DAFormer | DAFormer [23] | 7.3 | 75.1 | 51.7 | 48.0 | 15.1 | 39.4 | 15.3 | 51.6 | 33.6 | 27.8 | 38.5 | 4.0 | 28.5 |
| | MIC [25] | 10.8 | 76.4 | 53.3 | 52.7 | 16.0 | 41.8 | 20.7 | 51.9 | 13.3 | 55.2 | 44.8 | 9.3 | 32.5 |
| | CROVIA [50] | 16.3 | 75.1 | 59.6 | 60.0 | 19.1 | 46.0 | 20.5 | 56.1 | 37.6 | 50.7 | 45.3 | 10.9 | 36.8 |
| | **EAGLE** | **30.6** | **75.3** | **59.7** | **63.1** | **25.3** | **50.8** | **23.9** | **65.0** | **38.5** | **53.5** | **49.3** | **14.1** | **40.7** |
| | Supervised | 78.0 | 91.2 | 79.7 | 77.5 | 44.2 | 74.1 | 79.0 | 92.8 | 81.9 | 78.4 | 70.3 | 45.7 | 74.7 |
| Mask2Former | **EAGLE** | **31.4** | **75.1** | **57.5** | **59.2** | **19.5** | **48.6** | **22.9** | **64.6** | **37.8** | **52.8** | **48.5** | **13.8** | **40.1** |
| | Supervised | 75.5 | 91.6 | 79.1 | 77.7 | 42.1 | 73.2 | 76.8 | 91.8 | 81.1 | 77.6 | 67.8 | 43.4 | 73.1 |

mIoU results of our approach are higher than CROVIA [50] and MIC [25] by +4.8% and +9.0%. The IoU result of each class also consistently outperformed the prior methods. Highlighted that although our approach does NOT use depth labels, our results still outperform the one using depths, i.e., DADA [59]. It has emphasized that our approach is able to better capture the cross-view structural changes compared to prior methods. Figure 4 illustrates our qualitative results compared to ProDA [69] and CROVIA [50].

**GTA → UAVID** As shown in Table 4, our effectiveness outperforms prior polar view transformation [45] and domain adaptation approaches when measured by both mIoU performance and the IoU accuracy of each class. In particular, our mIoU performance using DeepLab and DAFormer network achieves 36.7% and 40.7%, respectively. Our results have substantially closed the performance gap with the supervised results. By using the better segmentation-based network, i.e., Mask2Former with ResNet, the performance of our approach is further improved to 40.1% compared to DeepLab.

## 4.4 Comparisons with Open-vocab Segmentation

We compare EAGLE with the prior open-vocab segmentation methods, i.e., DenseCLIP [40] and an adaptive prompting FreeSeg [38] with four settings, i.e., Source Only, with AdvEnt [58], and with SAC [1], and our Cross-View Adaptation in Eqn. (6) (without view-condition).

**Open-vocab Semantic Segmentation** As in Table 5, the mIoU performance of our proposed approach with cross-view adaptation outperforms prior DenseCLIP by a large margin on SYNTHIA → UAVID. By using our cross-view geometric adaptation loss, the performance of DenseCLIP and FreeSeg is further enhanced, i.e., higher than DenseCLIP and FreeSeg with SAC by +3.7% and +5.0%. While FreeSeg [38] with our cross-view adaptation slightly outperforms EAGLE due to its adaptive prompting, our EAGLE approach with the better view-condition prompting achieves higher mIoU

Table 5: Comparisons with Open-vocab Semantic Segmentation.

| | Method | SYNTHIA → UAVID | | | | | | GTA → UAVID | | | | | | |
|---|---|---|---|---|---|---|---|---|---|---|---|---|---|---|
| | | Road | Building | Car | Tree | Person | mIoU | Road | Building | Car | Tree | Terrain | Person | mIoU |
| FPN ResNet | DenseCLIP | 14.6 | 27.2 | 14.7 | 32.6 | 7.1 | 19.2 | 16.1 | 26.0 | 6.4 | 28.3 | 20.8 | 5.9 | 17.3 |
| | DenseCLIP + AdvEnt | 27.7 | 62.0 | 48.6 | 40.2 | 18.1 | 39.3 | 25.5 | 39.4 | 20.6 | 41.4 | 38.7 | 14.9 | 30.1 |
| | DenseCLIP + SAC | 28.6 | 63.5 | 51.5 | 43.4 | 18.3 | 41.1 | 17.2 | 52.3 | 30.8 | 35.7 | 41.9 | 15.3 | 32.2 |
| | DenseCLIP + Cross-View | 32.4 | 67.0 | 55.3 | 50.2 | 19.6 | 44.9 | 19.6 | 58.7 | 33.9 | 41.5 | 43.9 | 16.2 | 35.6 |
| FPN ViT | DenseCLIP | 17.2 | 28.9 | 16.9 | 37.3 | 8.6 | 21.8 | 17.7 | 28.3 | 8.9 | 33.1 | 23.5 | 6.3 | 19.6 |
| | DenseCLIP + AdvEnt | 28.1 | 67.0 | 49.9 | 39.8 | 17.2 | 40.4 | 16.5 | 51.3 | 29.8 | 33.9 | 41.0 | 15.2 | 31.3 |
| | DenseCLIP + SAC | 29.1 | 67.4 | 51.6 | 44.4 | 17.8 | 42.1 | 17.9 | 53.9 | 32.5 | 37.8 | 42.7 | 15.5 | 33.4 |
| | DenseCLIP + Cross-View | 31.6 | 71.4 | 53.9 | 50.1 | 21.9 | 45.8 | 20.6 | 60.8 | 35.8 | 45.0 | 44.6 | 16.8 | 37.3 |
| Mask2Former | FreeSeg | 18.4 | 30.0 | 17.9 | 41.5 | 8.9 | 23.4 | 18.0 | 28.7 | 9.8 | 33.9 | 24.0 | 6.3 | 20.1 |
| | FreeSeg + AdvEnt | 30.0 | 71.2 | 54.0 | 43.3 | 18.0 | 43.3 | 20.3 | 60.6 | 35.6 | 42.3 | 44.7 | 16.6 | 36.7 |
| | FreeSeg + SAC | 32.0 | 73.3 | 56.6 | 50.4 | 19.2 | 46.3 | 22.1 | 62.5 | 38.1 | 45.7 | 45.6 | 17.4 | 38.6 |
| | FreeSeg + Cross-View | 36.4 | 76.5 | 60.6 | 60.5 | 22.6 | 51.3 | 25.7 | 66.8 | 43.1 | 57.2 | 47.5 | 26.2 | 44.4 |
| | EAGLE | 36.8 | 75.5 | 61.3 | 60.8 | 21.2 | 51.1 | 27.3 | 66.8 | 42.3 | 35.5 | 47.1 | 25.1 | 44.0 |
| | EAGLE + View Condition | 38.4 | 76.1 | 62.8 | 62.1 | 21.8 | 52.2 | 29.2 | 67.1 | 45.2 | 56.6 | 48.5 | 27.9 | 45.7 |
| | Supervised | 79.8 | 92.6 | 82.9 | 79.1 | 48.0 | 76.5 | 80.5 | 93.3 | 82.7 | 79.2 | 71.3 | 49.9 | 76.1 |

Table 6: Comparisons with Open-vocab Segmentation on Seen (mIoU$^S$) and Unseen (mIoU$^U$) Classes.

| | Method | SYNTHIA → UAVID | | GTA → UAVID | |
|---|---|---|---|---|---|
| | | mIoU$^S$ | mIoU$^U$ | mIoU$^S$ | mIoU$^U$ |
| FPN ResNet | DenseCLIP + AdvEnt | 54.7 | 30.4 | 40.9 | 30.3 |
| | DenseCLIP + SAC | 56.2 | 32.1 | 44.1 | 31.4 |
| | DenseCLIP + Cross-View | 58.3 | 35.1 | 46.3 | 34.1 |
| FPN ViT | DenseCLIP + AdvEnt | 55.2 | 31.2 | 44.5 | 31.7 |
| | DenseCLIP + SAC | 56.5 | 33.2 | 46.1 | 33.4 |
| | DenseCLIP + Cross-View | 59.0 | 36.6 | 48.5 | 35.6 |
| Mask2Former | FreeSeg + AdvEnt | 55.8 | 31.1 | 46.5 | 33.5 |
| | FreeSeg + SAC | 58.0 | 34.8 | 48.6 | 36.1 |
| | FreeSeg + Cross-View | 60.2 | 38.3 | 50.7 | 38.2 |
| | EAGLE | 60.6 | 37.7 | 50.5 | 37.5 |
| | EAGLE + View Condition | 61.6 | 39.3 | 51.4 | 39.6 |
| | Fully Supervised | 85.1 | 63.6 | 81.9 | 64.5 |

performance. Similarly, our proposed cross-view loss consistently improves the performance of DenseCLIP and FreeSeg on GTA → UAVID. The mIoU results of DenseCLIP and FreeSeg using our cross-view loss achieve 37.3% and 44.4%. By further using the view-condition prompting mechanism, our mIoU result is considerably higher than FreeSeg with our cross-view adaptation by +1.3%. Figure 6 visualizes our qualitative results of our proposed approach.

**Open-vocab Segmentation on Unseen Classes** Table 6 illustrates the experimental results of our cross-view adaptation approach on unseen classes. In this experiment, we consider classes of Tree and Person as the unseen classes. As shown in the results, our cross-view adaptation approach with a view-condition prompting mechanism has achieved the best mIoU performance on unseen classes on both benchmarks, i.e., 39.3% and 39.6% on two benchmarks. Our experimental results have further confirmed the effectiveness and the generalizability of our cross-view geometric modeling and view-condition prompting approach to the open-vocab segmentation across views.

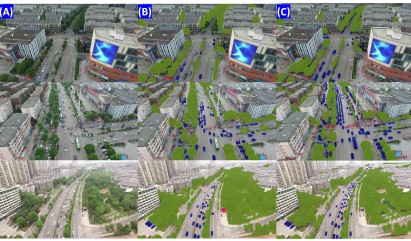

Figure 6: Results of Segmenting Cars, Trees, Persons. (A) Input, (B) FreeSeg [38], and (C) Our EAGLE.

**Real-to-Real Cross-view Adaptation Setting** We evaluated our approach in the real-to-real setting, i.e., BDD → UAVID. Our approach is evaluated in two different settings, i.e., Unsupervised Domain Adaptation and Open-Vocab Semantic Segmentation. As shown in Table 7, our results have shown a significant improvement in our approach in real-to-real settings in both unsupervised domain adaptation and open-vocab semantic segmentation. While the results of prior unsupervised domain adaptation, i.e.,

Table 7: Comparison with Prior Adaptation Methods and Open-Vocab Segmentation on Real-to-Real Cross-View Setting.

| Setting | Method | BDD → UAVID | | | | | | |
|---|---|---|---|---|---|---|---|---|
| | | Road | Building | Car | Tree | Terrain | Person | mIoU |
| Unsupervised Domain Adaptation | No Adaptation | 19.2 | 8.5 | 34.6 | 18.4 | 13.6 | 4.0 | 16.4 |
| | BiMaL [51] | 19.5 | 52.4 | 35.1 | 50.4 | 46.0 | 10.2 | 35.6 |
| | Polar Trans. [45] | 21.1 | 9.6 | 36.4 | 24.1 | 14.6 | 4.6 | 18.4 |
| | EAGLE (DeepLab) | 24.0 | 53.8 | 39.0 | 52.2 | 48.3 | 16.9 | 39.0 |
| | DAFormer [23] | 25.8 | 65.4 | 38.7 | 54.5 | 51.3 | 14.8 | 41.8 |
| | EAGLE (DAFormer) | 29.0 | 66.1 | 41.5 | 55.6 | 53.3 | 21.5 | 44.5 |
| Open-Vocab Seg | DenseCLIP + Cross-View | 25.9 | 60.9 | 39.5 | 35.5 | 47.1 | 33.9 | 40.5 |
| | FreeSeg + Cross-View | 32.6 | 67.3 | 47.9 | 51.8 | 50.3 | 37.2 | 47.9 |
| | EAGLE | 35.4 | 68.9 | 50.6 | 59.2 | 51.7 | 38.6 | 50.7 |

BiMaL [51] and DAFormer [23], gain limited performance due to their limits in cross-view learning, our method outperforms other methods these prior methods by a large margin.

## 5 Conclusions

This paper has presented a novel unsupervised cross-view adaptation approach that models the geometric correlation across views. We have introduced the Geodesic Flow-based metric to better model geometric structural changes across camera views. In addition, a new view-condition prompting mechanism has been presented to further improve the cross-view modeling. Through our theoretical analysis and SOTA performance on both unsupervised cross-view adaptation and open-vocab segmentation, our approach has shown its effectiveness in cross-view modeling and improved robustness of segmentation models across views.

**Limitations** Our study has selected a set of learning hyper-parameters to support our hypothesis and experiments. However, this work can potentially contain several limitations related to learning parameters and linear relation hypothesis in Eqn. (4). The details of the limitations are discussed in the appendix. We believe that these limitations will motivate future studies to improve our unsupervised cross-view adaptation learning approach.

**Acknowledgment** This work is partly supported by NSF Data Science, Data Analytics that are Robust and Trusted (DART), NSF SBIR Phase 2, and Arkansas Biosciences Institute (ABI) grants. We also acknowledge the Arkansas High-Performance Computing Center for providing GPUs.

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

# Appendix

## 1 Proof of Eqn. (9)

As shown in our Eqn. (16), our Geodesic Flow-based metrics have the upper bound as follows:

$$\forall \mathbf{x}_s, \mathbf{x}_t: \quad \mathcal{D}_x(\mathbf{x}_s, \mathbf{x}_t) = 1 - \frac{\mathbf{x}_s^\top \mathbf{Q} \mathbf{x}_t}{||\mathbf{Q}^{1/2}\mathbf{x}_s||||\mathbf{Q}^{1/2}\mathbf{x}_t||} \leq 2$$
$$\forall \mathbf{y}_s, \mathbf{y}_t: \quad \mathcal{D}_y(\mathbf{y}_s, \mathbf{y}_t) = 1 - \frac{\mathbf{y}_s^\top \mathbf{Q} \mathbf{y}_t}{||\mathbf{Q}^{1/2}\mathbf{y}_s||||\mathbf{Q}^{1/2}\mathbf{y}_t||} \leq 2 \tag{16}$$

In addition, as $\mathcal{D}_x$ is the distance metric, this metric should satisfy the following triangular inequality as follows:

$$\mathcal{D}_x(\mathbf{x}_s, \bar{\mathbf{x}}_t) \leq \mathcal{D}_x(\mathbf{x}_s, \mathbf{x}_t) + \mathcal{D}_x(\mathbf{x}_t, \bar{\mathbf{x}}_t) \tag{17}$$

Similarly, $\mathcal{D}_y$ should satisfy the following triangular inequality as follows:

$$\begin{aligned}
& \mathcal{D}_y(\mathbf{y}_t, \bar{\mathbf{y}}_t) + \mathcal{D}_y(\bar{\mathbf{y}}_t, \mathbf{y}_s) \geq \mathcal{D}_y(\mathbf{y}_s, \mathbf{y}_t) \\
\Leftrightarrow\ & \mathcal{D}_y(\mathbf{y}_t, \bar{\mathbf{y}}_t) \geq \mathcal{D}_y(\mathbf{y}_s, \mathbf{y}_t) - \mathcal{D}_y(\bar{\mathbf{y}}_t, \mathbf{y}_s) \\
\Leftrightarrow\ & -\alpha \mathcal{D}_y(\mathbf{y}_t, \bar{\mathbf{y}}_t) \leq -\alpha \left( \mathcal{D}_y(\mathbf{y}_s, \mathbf{y}_t) - \mathcal{D}_y(\bar{\mathbf{y}}_t, \mathbf{y}_s) \right)
\end{aligned} \tag{18}$$

Then, from Eqn. (16) and Eqn. (17) above, we can further derive as follows:

$$\begin{aligned}
& \mathcal{D}_x(\mathbf{x}_s, \bar{\mathbf{x}}_t) - \alpha \mathcal{D}_y(\mathbf{y}_s, \bar{\mathbf{y}}_t) \\
& \leq \mathcal{D}_x(\mathbf{x}_s, \mathbf{x}_t) + \mathcal{D}_x(\mathbf{x}_t, \bar{\mathbf{x}}_t) - \alpha \left( \mathcal{D}_y(\mathbf{y}_s, \mathbf{y}_t) - \mathcal{D}_y(\bar{\mathbf{y}}_t, \mathbf{y}_s) \right) \\
& \leq \mathcal{D}_x(\mathbf{x}_s, \mathbf{x}_t) - \alpha \mathcal{D}_y(\mathbf{y}_s, \mathbf{y}_t) + \mathcal{D}_x(\mathbf{x}_t, \bar{\mathbf{x}}_t) + \alpha \mathcal{D}_y(\bar{\mathbf{y}}_t, \mathbf{y}_s) \\
& \leq \mathcal{D}_x(\mathbf{x}_s, \mathbf{x}_t) - \alpha \mathcal{D}_y(\mathbf{y}_s, \mathbf{y}_t) + \underbrace{2(1 + \alpha)}_{Constant}
\end{aligned} \tag{19}$$

Since $\alpha$ is the constant linear scale value, therefore, we can further derive as follows:

$$\begin{aligned}
\Rightarrow\ & ||\mathcal{D}_x(\mathbf{x}_s, \bar{\mathbf{x}}_t) - \alpha \mathcal{D}_y(\mathbf{y}_s, \bar{\mathbf{y}}_t)|| \\
& = \mathcal{O}(||\mathcal{D}_x(\mathbf{x}_s, \mathbf{x}_t) - \alpha \mathcal{D}_y(\mathbf{y}_s, \mathbf{y}_t)||)
\end{aligned} \tag{20}$$

## 2 Implementation

We follow the implementation of Mask2Former [8] and FreeSeg [38] with ResNet [20] and Swin backbones [32] for our segmentation network. In particular, we adopt Mask2Former with Semantic Context Interaction of FreeSeg [38] for our open-vocab segmentation network. We use the pre-trained text encoder of CLIP [39]. The textual features $\mathbf{f}_s^p$ and $\mathbf{f}_t^p$ are obtained by the CLIP textual encoder. Following common practices [38, 31], we adopt the open-vocab segmentation loss of FreeSeg [38] to our supervised loss $\mathcal{L}_{Mask}$. For experiments without prompting, we use the Mask2Former network. Following the UAV protocol of [61], the image size is set to $512 \times 512$. The linear scale factors $\alpha$ and $\gamma$ are set to $\alpha = 1.5$ and $\gamma = 1.0$, respectively. For the Geodesic Flow modeling, we adopt the implementation of generalized SVD decomposition [18, 47] in the framework. The subspace dimension in our geodesic flow-based metrics is set to $D = 256$. The batch size and the base learning rate in our experiments are set to 16 and $2.5 \times 10^{-4}$. The balanced weights of losses in our experiments are set to $\lambda_I = 1.0$ and $\lambda_P = 0.5$. During training, the classes in the prompts are generated similarly for both view images.

In our Geodesic Flow-based metrics, the subspaces of images and ground-truth segmentation of the source domain are pre-computed on the entire data. For the language space, we compute the subspaces of each view based on the textual feature representations of all possible prompts in each domain. Meanwhile, the subspaces of the segmentation on the target domain are computed based on the current batch of training. For the implementation of DenseCLIP [40] and FreeSeg [38] with AdvEnt [58], we perform the adaptation process on the mask predictions. Meanwhile, we adopt the pseudo labels and the self-supervised framework of SAC[1] for the implementation of DenseCLIP [40] and FreeSeg [38] with SAC [1].

Table 8: Effectiveness of Batch Size.

| Batch Size | SYNTHIA → UAVID | | | | | | GTA → UAVID | | | | | | |
|---|---|---|---|---|---|---|---|---|---|---|---|---|---|
| | Road | Building | Car | Tree | Person | mIoU | Road | Building | Car | Tree | Terrain | Person | mIoU |
| 4 | 25.5 | 58.9 | 46.5 | 29.2 | 15.9 | 35.2 | 17.6 | 50.6 | 29.7 | 26.0 | 41.1 | 22.4 | 31.2 |
| 8 | 35.3 | 68.9 | 57.6 | 54.4 | 20.9 | 47.4 | 25.1 | 57.6 | 39.3 | 45.5 | 44.9 | 26.2 | 39.8 |
| 16 | **38.4** | **76.1** | **62.8** | **62.1** | **21.8** | **52.2** | **29.2** | **67.1** | **45.2** | **56.6** | **48.5** | **27.9** | **45.7** |

## 3 Ablation Study

**Effectiveness of Batch Size** Table 8 illustrates the impact of the batch size on the performance of cross-view domain adaptation. By increasing the batch size, the mIoU performance is also increased accordingly on both benchmarks. This result has illustrated that the small batch size could not have enough samples to approximate the subspace that represents geometric structures. Meanwhile, the subspace created from the large batch size will be ale to capture the geometric structure of drone-view scenes. However, due to the limitation of GPU resources, we could not evaluate the cross-view adaptation model with larger batch size.

**Subspace Representation of Geodesic Flow-based Metrics** To illustrate the ability of structural learning of our geodesic flow-based metrics, we use a subset of images of the car-view and the drone-view dataset to visualize the base structure of subspaces obtained from the PCA algorithm. Fig. 7 visualizes the mean structures of car-view and drone-view images. As shown in Fig. 7, The subspaces of car-view images represent the geometric structures of car-view data, i.e., the road in the middle, buildings, trees on two sides, etc. Meanwhile, the geometric structures of the drone view have also been illustrated in the figure with structures and topological distributions of objects (e.g., the road in the middle and trees and buildings on the sides) on the scenes. The results have illustrated the base geometric structures of the car-view and the drone-view data. Then, by modeling the geodesic flow path across two subspaces, our metric is able to measure the cross-view geometric structural changes (i.e., the change of structures and topological layouts of the scene) from the car view to the drone view. Our experimental results in other ablation studies have further confirmed our effectiveness in geometric structural modeling across views. Figure 8 illustrates the feature distributions with and without our proposed approach. As shown in Figure 8, our approach can help to improve the feature representations of classes, and the cluster of each class is more compact, especially in classes of car, tree, and person.

## 4 Discussion of Limitations and Broader Impact

**Limitations.** In our paper, we have specified a set of hyper-parameters and network designs to support our hypothesis and theoretical analysis. However, our proposed approach could potentially consist of several limitations. First, our work focuses on studying the impact of cross-view geometric adaptation loss and view-condition prompting mechanisms on the segmentation models across views. The balanced weights among weights, i.e., $\lambda_I$ and $\lambda_P$, have not been fully exploited. We leave this investigation as our future experiments. Second, although the datasets and benchmarks used in our

**Structure of Car-View Image** | **Structure of Drone-View Image**

Figure 7: The Structures of Subspaces of Car-View and Drone-View Dataset Learned From a Subset of Images.

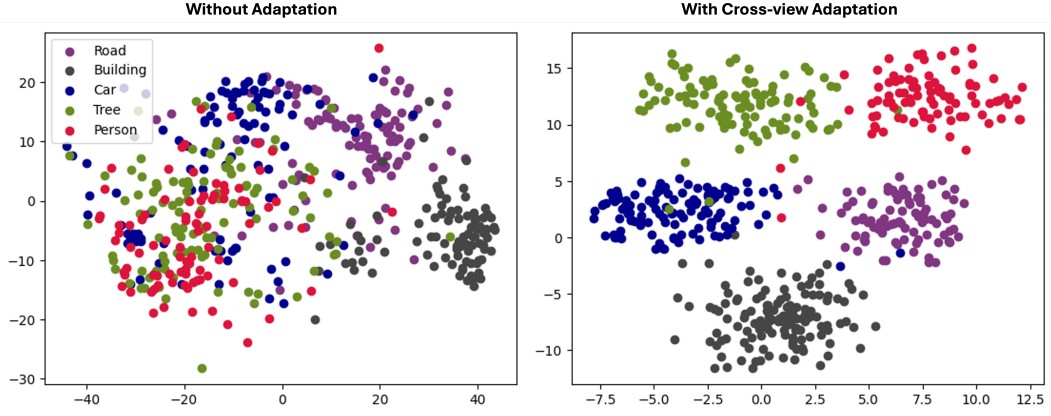

Figure 8: The Feature Distribution of Classes in SYNTHIA → UAVID Experiments.

experiments have sufficiently illustrated the effectiveness of our proposed cross-view adaptation learning approach, the lack of diverse classes and categories in datasets is also a potential limitation. Third, the hypothesis of the linear relations across views of images and segmentation mask, i.e., $\alpha$, and textual representations and segmentation masks, i.e., $\gamma$, could limit the performance of the relation. The non-trivial relations across views should be deeply exploited in future research. Also, while the implementation of Mask2Former and FreeSeg is adopted to develop our approach, the experiments with other open-vocab segmentation networks should be considered in subsequent research studies. These aforementioned limitations will motivate new studies to further improve the methodology, datasets, and benchmarks of the cross-view adaptation learning paradigm.

**Broader Impact.** Our paper could bring significant potential for various applications that require learning across camera viewpoints. Our approach enables generalizability across camera views, thus enhancing the robustness of the segmentation model across views. In addition, our approach helps to reuse off-the-shelf large-scale data while reducing the effort of manually labeling data of new camera views.

## 5 Other Related Work

While the important and closely related work to our approach has been presented in our main paper, we also would like to review some other research studies that are related to our method as follows. In particular, Brady et al. [72] presented a cross-view transformer that learns the camera-aware positional embeddings. Although the views are captured from left and right angles, the camera positions in the approach remain at the same altitude. Similarly, Pan et al. [36] present a View Parsing Network to accumulate features across first-view observations with multiple angles. Yao et al. [66] proposed a semi-supervised learning approach to learn the segmentation model from multiple views of an image. Huang et al. [26] a cross-style regularization for domain adaptation in panoptic segmentation by imposing the consistency of the segmentation between the target images and stylized target images. Wang et al. [62] proposed a viewpoint adaptation framework for the person re-identification problem by using the generative model to generate training data across various viewpoints. Hou et al. [22] presented a matching cross-domain data approach to domain adaptation in visual classification. Sun et al. [48] proposed a cross-view facial expression adaptation framework to parallel synthesize and recognize cross-view facial expressions. Goyal et al. [19] introduced a cross-view action recognition approach to transferring the feature representations to different views. Zhang et al. [70] proposed a multi-view crowd counting approach that adaptively chooses and aggregates multi-cameras and a noise view regularization. Armando et al. [2] proposed a self-supervised pre-training approach to human understanding learned on pairs of images captured from different viewpoints. Then, the pre-trained models are later used for various downstream human-centric tasks. In summary, these prior cross-view methods could require either a pair of cross-view images [2] or images captured at the same altitude with different angles [26, 72, 22]. In addition, the cross-view geometric correlation modeling has not been exploited in these prior studies [26, 72, 22, 2]

