# OpenReview forum: "EAGLE: Efficient Adaptive Geometry-based Learning in Cross-view Understanding"
_NeurIPS.cc/2024/Conference — NeurIPS 2024 poster_

### Official Review · Reviewer_29y1 · 2024-07-09

**Soundness:** 2
**Presentation:** 3
**Contribution:** 2
**Rating:** 3
**Confidence:** 5

**Summary:**

This paper proposes a domain adaptive semantic segmentation method under the cross-view (front view to top view) setting. It addresses this using vision-language models for additional supervision. It proposes a cross-view geometric constraint to model the structural changes and similarities between two vastly different viewpoint. The paper presents detailed quantitative results, with comparisons and ablations

**Strengths:**

1. The paper addresses an important and practical problem.

2. The method proposed in the paper is well-tailored for the problem at hand, and appears to be solid. It seems intuitive and logical.

3. The quantitative results presented clearly show the efficacy of the method.

**Weaknesses:**

My main problem with the paper is the lack of qualitative results. The main paper barely has any qualitative results. The supplementary video shows just one specific video. I am unable to judge the performance of the method without good qualitative analysis.

As in all semantic segmentation papers, it would be good to see where prior work fails, and how each of the components presented in the method help the case. Essentially, it is important to provide qualitative results for the comparisons, the ablations, and as well as the method.

Given that the performance improvement as depicted by the quantitative results is large, the qualitative improvements should be clearly visible too, and hence it is beneficial to add those results.

Also, one main motivation for using a VLM is the open set setting (as described in the abstract). While described as a key point in the paper, none of the experimental benchmarks deal with that.

**Questions:**

Since it is not possible to provide qualitative results in the rebuttal, and since it not possible to properly validate the effectiveness of the proposed semantic segmentation method without qualitative results, I am rejecting the paper at this stage.

**Limitations:**

Yes, in page 9

---

> ### Author Rebuttal · Authors · 2024-08-05
>
> Dear Reviewer 29y1,
>
>
> We would like to express our gratitude for your careful reading and valuable feedback.
> We are very happy you encourage that ***our paper addresses an important and practical problem, our proposed approach is well-tailored for the problem and appears to be solid and logical, and our approach also achieves solid quantitative performance***.
> In addition, we would like to emphasize that other reviewers have also encouraged ***our quantitative results in our experiments and ablation studies are well-designed to illustrate the effectiveness of our proposed approach*** (Reviewer 8Etw, upJj, and JhPV). We appreciate your constructive comments and would like to address these points as follows.
>
>
> [Q1] **Lack of qualitative results**
>
>
> [A1] We have included the qualitative results of our ablation study in the pdf file of our rebuttal. In the model without prompting, Figure 1 illustrates the results of our cross-view adaptation compared to without adaptation. As shown in the results, our approach can effectively segment the objects in the drone view. We also compare with the prior ProDA [1] method in this figure. Our qualitative results remain better than the prior adaptation method. For the model with prompting, Figure 2 illustrates the effectiveness of our approach in 3 cases, without adaptation, with cross-view adaptation, and with view-condition prompting. As shown in the results, our cross-view adaptation can efficiently model the segmentation of the view. By using the view-condition prompting, our model can further improve the segmentation of persons and vehicles. The images are better viewed in color and 2x zoom. In addition to the quantitative results in Table 2, our qualitative results further confirm the effectiveness of our proposed approach. **We will release our implementation for the reproducibility of both quantitative and qualitative results.** We will release more qualitative results and comparisons in the final version of our paper.
>
>
>
>
>
>
> [Q2] **Using a VLM is the open-set setting**
>
>
> [A2] In Tables 5-6 in our paper, we report the performance of open-vocab segmentation (i.e. DenseCLIP, FreSeg). Specifically, The DenseCLIP model is developed based on the vision-language model (i.e., CLIP). These experimental results show the limited performance of these methods in the cross-view settings. The results also illustrate the effectiveness of our proposed approach in an open-vocab segmentation setting.
> In Table 6, we also present our experimental results in the open-set setting where classes of 'Tree' and 'Person' are considered as the unseen classes (which are not used during training).
> We would like to highlight that other reviewers encourage ***our experiments and ablation studies are well-designed for contributions*** (Reviewer 8Etw, upJj, and JhPV) and ***our experimental results are solid*** (Reviewers upJj, 8Etw, and JhPV).
>
> References
>
> [1] P Zhang, et al. Prototypical pseudo label denoising and target structure learning for domain adaptive semantic segmentation. CVPR, 2021.

---

> > ### Comment · Reviewer_29y1 · 2024-08-09
> > **Response to rebuttal**
> >
> > Thanks a lot for the detailed rebuttal, I really appreciate your time and efforts in this.
> >
> > I went through the other reviews, as well as the detailed responses.
> >
> > Thank you for the clarification on the open-set setting.
> >
> > Regarding qualitative results, is there any particular reason why comparisons are shown against PRODA (2021) and not CROVIA (2023)?
> >
> > I see in Table 4 that the improvement of the method in the paper over CROVIA is 5%, so it would be nice to see where exactly the improvement is coming from. I understand that the code of CROVIA is not open-source, but since the NeurIPS submission has additional results on CROVIA (over what is reported in the CROVIA paper at https://arxiv.org/pdf/2304.07199), I believe you have an implementation, it'll be nice to see those qualitative comparisons.

---

> > > ### Author Response · Authors · 2024-08-09
> > > **Response to Reviewer's Feedback**
> > >
> > > Dear Reviewer 29y1,
> > >
> > > We are glad that our rebuttal has addressed your concerns related to open-set settings.
> > >
> > > For clarification of our visualization results, we prefer the ProDA method for comparison in our rebuttal since ProDA is officially published in CVPR 2021. Meanwhile, CROVIA is only available in a preprint in arXiv. We would like to thank you for your suggestions. We will add more qualitative comparisons between our method and CROVIA in our revised paper.
> > >
> > > For clarification of performance improvement compared to CROVIA, our EAGLE approach models the cross-view geometric structural changes via the geodesic flow path. Meanwhile, CROVIA measures the cross-view structural changes by measuring the distribution shift. Thus, our approach can efficiently measure structural changes across views via their manifold structure between two views, more intuitive compared to measuring the distribution shift used in CROVIA. For example, as shown in Figure 5 (page 9) in the CROVIA paper (https://arxiv.org/pdf/2304.07199), we have seen that the segmentation of the class “Person” is not as good as our EAGLE approach. Our quantitative performance in Table 4 also illustrates that our approach performs better than CROVIA.
> > >
> > > If you have any other concerns, please do not hesitate to raise your questions. We are happy to address your questions.
> > >
> > > Thank you very much,
> > >
> > > Authors

---

> > > > ### Author Response · Authors · 2024-08-13
> > > > **Rebuttal Follow Up**
> > > >
> > > > Dear Reviewer 29y1,
> > > >
> > > > Thank you very much for your insightful feedback.
> > > >
> > > > The reviewer-author discussion deadline is nearing. We have not received your final response to our final rebuttal. Therefore, we are reaching out to you to ensure that our rebuttal effectively addresses your concerns. If you have any further questions, please let us know. We appreciate your invaluable input.
> > > >
> > > > Thank you very much,
> > > >
> > > > Authors

---

> > > > > ### Comment · Reviewer_29y1 · 2024-08-13
> > > > >
> > > > > Dear authors,
> > > > >
> > > > > Thank you for your responses.
> > > > >
> > > > > I believe it'll be great to see qualitative comparisons against CROVIA.
> > > > >
> > > > > I feel these comparisons will be significant in demonstrating the superiority of the method given the large quantitative improvements. It is unclear why these comparisons were omitted in the rebuttal pdf, especially considering that ProDA is a 2021 paper. Additionally, the NeurIPS submission provided further quantitative results on CROVIA (2023) beyond those in the original CROVIA paper, suggesting that an implementation was available.
> > > > >
> > > > > Thanks,
> > > > > Reviewer

---

> ### Author Response · Authors · 2024-08-13
> **Feedback to Reviewer Response**
>
> Dear Reviewer 29y1,
>
>
> As shown in the following table (the detailed comparison can be found in Table 4, page 8 in the paper), our EAGLE has outperformed CROVIA and ProDA. We believe the improvement compared to CROVIA is also illustrated in the visualization (as our comparison with ProDA) due to our reproduced results.  **In our revised paper, we will add more qualitative comparisons between our method and CROVIA and release our implementation for research reproducibility (both quantitative and qualitative results).**
>
> | SYNTHIA to UAVID | Road | Building | Car | Tree | Person | mIoU |
> |---|---|---|---|---|---|---|
> | ProDA | 10.6 | 64.7 | 34.1 | 44.5 | 17.0 | 34.2 |
> | CROVIA | 10.6 | 65.7 | 51.7 | 55.6 | 17.0 | 40.0 |
> | **EAGLE** | **29.9** | **65.7** | **55.5** | **56.8** | **18.3** | **45.2** |
>
> If you have any other concerns, please do not hesitate to raise your questions. We are happy to address your questions.
>
>
> Thank you very much,
>
> Authors

---

### Official Review · Reviewer_JhPV · 2024-07-11

**Soundness:** 4
**Presentation:** 3
**Contribution:** 4
**Rating:** 7
**Confidence:** 4

**Summary:**

The paper introduces a novel method for Unsupervised Domain Adaption to adapt an open-vocabulary segmentation model across different views. To achieve this, the authors introduce a cross-view geometric constraint that captures structural changes between different views. Further, a Geodesic Flow-based Metric is introduced to measure the structural changes across scenes. They also adapt the prompting scheme to take into account the change in viewpoints.

**Strengths:**

- The paper feature a good range of contributions, which are intuitive and interesting
- The proposed task of unsupervised cross-view adaptation is novel and seems useful
- The idea is well motivated
- The paper is well written and clearly explains the proposed concepts
- Table 2 nicely outlines the impact of the individual technical contributions

**Weaknesses:**

- The layout of the paper seems a bit cramped. It would have been nice to have larger visualizations. Also, many of the tables, especially those in the ablations, seem super cramped and the reader has to zoom in significantly. I understand the authors wanted to put as much information as possible into the paper, but this led to a presentation that is not top notch.
- Table 3 has no visual separation between the results for different classes, and overall mIoU. This visual separation (e.g. vertical line) would be nice to have.

**Questions:**

Personally, I would appreciate if the authors could further improve the presentation of the paper. While the technical contributions are solid and the writing is nice, the visualizations and tables would benefit from being larger and better integrated with the text. For example, by moving Table 6 to the supplementary, the authors could free up space for the very relevant Table 5. I would appreciate an improved presentation, but also understand that this can be challenging with the given space constraints.

**Limitations:**

The authors have sufficiently addressed the limitations of their approach.

---

> ### Author Rebuttal · Authors · 2024-08-05
>
> Dear Reviewer JhPV,
>
>
> We greatly appreciate your insightful review and valuable feedback.
> We are very happy you encourage that ***our paper is well-written and features a good range of contributions, our problem is meaningful, and our idea is well-motivated***.
> We appreciate your constructive comments and would like to address these points as follows.
>
>
> [Q1] **Layout of Paper**
>
>
> [A1] Thank you very much for your feedback. We will update the layout organization of our paper for better readability.
>
>
> [Q2] **Visual Separation in Tables**
>
>
> [A2] Thank you very much for your feedback. Per your suggestions, we will add more space to the tables for better readability.
>
>
> [Q3] **Free Up Space and Improve Presentation**
>
>
> [A3] Thank you very much for your feedback. We will update our paper according to your suggestions to improve our presentation and the readability.

---

> > ### Comment · Reviewer_JhPV · 2024-08-13
> > **Response to rebuttal**
> >
> > Dear authors,
> > Thanks for addressing my concerns.
> > I will keep my rating as I remain convinced that this paper is a candidate for acceptance.
> >
> > Best regards

---

> > > ### Author Response · Authors · 2024-08-13
> > > **Response to Reviewer Feedback**
> > >
> > > Dear Reviewer JhPV,
> > >
> > > Thank you for your invaluable feedback and positive rating. We're pleased that our rebuttal has addressed your concerns. We are dedicated to updating our paper based on your suggestion to improve our paper's presentation.
> > >
> > > Thank you very much,
> > >
> > > Authors

---

### Official Review · Reviewer_8Etw · 2024-07-12

**Soundness:** 3
**Presentation:** 3
**Contribution:** 3
**Rating:** 6
**Confidence:** 4

**Summary:**

This work proposed a novel unsupervised adaptation method for modeling structural change across different views. Additionally, the paper introduced a new metric for cross-view changes and a new prompting mechanism for cross-view open vocabulary segmentation. Through extensive experiments, the paper shows SoTA performance compared to previous unsupervised methods in cross-view semantic scene understanding.

**Strengths:**

1.The proposed method can be trained on unpaired data, enhancing its potential for practical applications.
2.The paper proposed a new cross-view change modeling method via geodesic flow path, which effectively models the cross-view segmentation correlation.
3.The newly introduced method demonstrated SoTA performance across various cross-view adaptation benchmarks.
4.The paper provides thorough mathematical derivations and theoretical analysis, along with well-designed experiments.

**Weaknesses:**

1.The visualization results in this paper are limited. It is necessary to supplement more visualizations of segmentation results compared with the existing methods in the paper, which will help support the experimental conclusions.
2.Generalizing the experiments to include more UAD datasets, such as SynDrone[1], UDD[2], and ICG Drone[3], would enhance the robustness and credibility of the results.
3.The motivation behind the design of the adaptation loss should be explained more clearly.
4.The performance improvement in open-vocab segmentation is not very significant.

[1] SynDrone – Multi-modal UAV Dataset for Urban Scenarios.
[2] Large-scale structure from motion with semantic constraints of aerial images.
[3] ICG Drone Dataset.

**Questions:**

1.The performance of the building category in open-vocab segmentation on unseen classes is intriguing.
2.The notation "ν ∈ [0..1] → Π(ν)" in line 241 is misleading.

**Limitations:**

The authors discussed the limitation of this paper in the appendix, including the choice of hyperparameters, lack of more diverse class labels and flaws in their original mathematical hypothesis.

---

> ### Author Rebuttal · Authors · 2024-08-05
>
> Dear Reviewer 8Etw,
>
>
> We are grateful for your careful reading and constructive feedback.
> We appreciate your highlighting that ***our proposed approach is efficient and well-designed, our problem is practical, and the method achieves solid performance***.
> We appreciate your constructive comments and would like to address these points as follows.
>
>
> [Q1] **Additional Visualization Results**
>
>
> [A1] To further illustrate our qualitative results, we add more visualization results (Figure 1 and Figure 2) in the pdf file of our rebuttal. In addition, we will release more qualitative results in the supplementary of our paper. We will also release our implementation for the reproducibility of both quantitative and qualitative results.
>
>
> [Q2] **Generalizing the experiments to include more UAV datasets**
>
>
> [A2]  We chose UAVID since these datasets have a great overlap with the source dataset (GTA, SYNTHIA, and BDD). We acknowledge the suggestion of the reviewer. Please refer to **[KP3]** for additional results of cross-view adaptation on UDD [1]. Due to the time limitation of the rebuttal period, we leave the experiments and investigation of other benchmarks of SynDrone [2] and ICG Drone [3] in our future work. It should be noted that we have also discussed the limitations of the dataset in the appendix.
>
>
>
>
> [Q3] **Performance of the building category in open-vocab segmentation on unseen classes**
>
>
> [A3] For clarification, in Table 5, all classes are used during training. In Table 6, we consider the classes of Tree and Person as the unseen classes. Therefore, the performance of building categories in our approach is valid and reasonable due to the effectiveness of our proposed method.
>
>
> [Q4] **Motivation behind the design of the adaptation loss**
>
>
> [A4] In our approach, we propose the adaptation loss by modeling the Cross-view Structural Change between source (car-view) and target (drone-view) domains. In particular, we first analyze the cross-view geometric correlation between two domains by analyzing the change of camera views and the equivalent transformation between image and segmentation output. Then, we propose to model the cross-view structural change via the geodesic flow. In particular, our loss measures the cross-view structural changes between the source and the target domains based on their manifold structures via the geodesic flow.
>
>
> [Q5] **The performance improvement in open-vocab segmentation is not very significant**
>
>
> [A5] We respectfully but strongly disagree with the reviewer on this point. As shown in Table 5, the performance of our open-cab experiment remains significant. For example, when we use our cross-view learning loss integrated with DenseCLIP (ViT) and FreSeg, the performance is up to 45.8% and  51.3 % on SYNTHIA $\to$ UAVID. Meanwhile, without our cross-view learning, the performance is only 21.8% and 23.4%.
> Our results still outperform prior adaptation methods (i.e., AdvEnt and SAC). Performance is even further improved when using our view-condition prompting.
>
>
> [Q4] **Misleading Notation**
>
>
> [A4] Thank you very much for your feedback. We will update our notation promptly.
>
>
> References
>
>
> [1] Y Chen, et al. Large-scale structure from motion with semantic constraints of aerial images. PRCV, 2018.
>
>
> [2] G Rizzoli, et al. SynDrone – Multi-modal UAV Dataset for Urban Scenarios. ICCVW, 2023.
>
>
> [3]  ICG Drone Dataset. http://dronedataset.icg.tugraz.at/.

---

> > ### Comment · Reviewer_8Etw · 2024-08-13
> >
> > I appreciate the authors for providing clarifications and additional experimental results, which have addressed some of my concerns. I will keep my score.

---

> > > ### Author Response · Authors · 2024-08-13
> > > **Response to Reviewer Feedback**
> > >
> > > Dear Reviewer 8Etw,
> > >
> > > We would like to thank you for your invaluable feedback and positive rating. We are glad that our rebuttal has addressed your concerns. We are committed to revising our paper according to your recommendations to improve the quality of our paper.
> > >
> > > Thank you very much,
> > >
> > > Authors

---

### Official Review · Reviewer_upJj · 2024-07-19

**Soundness:** 3
**Presentation:** 3
**Contribution:** 3
**Rating:** 6
**Confidence:** 4

**Summary:**

The paper tackles a problem called cross-view semantic segmentation by using unsupervised domain adaptation methods. The cross-view means from the front-view to top-down view, ie, from car to drone. It recognizes the limitations of existing unsupervised domain adaptation and open-vocabulary semantic segmentation methods in handling geometric variations across different camera views. The paper demonstrates the effectiveness of the proposed approach through extensive experiments on cross-view adaptation benchmarks, achieving state-of-the-art performance compared to existing UDA methods.

**Strengths:**

The authors propose a novel unsupervised cross-view adaptation approach. This approach includes a cross-view geometric constraint to model structural changes between images and segmentation masks, a geodesic flow-based correlation metric for efficient geometric comparison, and a view-condition prompting mechanism to enhance the capabilities.


The proposed method has significant gains as compared to the initial method CROVIA for car-to-drone scene segmentation UDA. Although this is not a new setting, the improvement compared to the baseline is good.

The experiments are conducted in three settings of the car-to-drone UDA, and it also includes ablation study to verify effect of different components.

**Weaknesses:**

The comparison between the other view transformation methods and the proposed cross-view learning method is not included.

Most of the UDA methods are tailored for the front-view street scenes, like AdvEnt, ProDA, SAC. More recent and advanced open-vocab semantic segmentation methods should be included in the experiment section. For example, x-decoder, cat-seg, clip-as-RNN, etc.

**Questions:**

How is the difference of UDA settings between CROVIA and the proposed in this paper?


In the introduction, the authors claim that “However, the open-vocab perception models remain unable to generalize across camera viewpoints.” It would be better to show the limitation or performance along with this observation. For example, the model with and without using a cross-view setting.


From eq 3 to 5, for the purpose of geometric adaptation on unpaired data, the problem is that the matrices between source and target domain data are not available. How about using estimated camera matrices for the geometric correlation in eq 3 and 4?


The cross-view learning framework is based on the geodesic flow. How about the comparison with other view transformation methods? For example, in the birds-eye-view semantic segmentation domain, there are different transformation methods, such as:
[1] Pan, B., Sun, J., Leung, H. Y. T., Andonian, A., & Zhou, B. (2020). Cross-view semantic segmentation for sensing surroundings. IEEE Robotics and Automation Letters, 5(3), 4867-4873.


In Table 4, how about the comparison for BDD→ UAVID? Also, how about applying tye cross-view setting for other DA methods that are proposed for front-view?


How about using the state-of-the-art open-vocab methods? For example, x-decoder, cat-seg, clip-as-RNN, to name a few.

[2] Zou, X., Dou, Z. Y., Yang, J., Gan, Z., Li, L., Li, C., ... & Gao, J. (2023). Generalized decoding for pixel, image, and language. In Proceedings of the IEEE/CVF Conference on Computer Vision and Pattern Recognition (pp. 15116-15127).

[3] Cho, S., Shin, H., Hong, S., Arnab, A., Seo, P. H., & Kim, S. (2024). Cat-seg: Cost aggregation for open-vocabulary semantic segmentation. In Proceedings of the IEEE/CVF Conference on Computer Vision and Pattern Recognition (pp. 4113-4123).

[4] Sun, S., Li, R., Torr, P., Gu, X., & Li, S. (2024). Clip as rnn: Segment countless visual concepts without training endeavor. In Proceedings of the IEEE/CVF Conference on Computer Vision and Pattern Recognition (pp. 13171-13182).

[5] Wysoczańska, M., Siméoni, O., Ramamonjisoa, M., Bursuc, A., Trzciński, T., & Pérez, P. (2023). Clip-dinoiser: Teaching clip a few dino tricks. arXiv preprint arXiv:2312.12359.


In Table 4, what is the reason that the method with DAFormer outperforms the one with Mask2Former. It is also very interesting why applying a larger or advanced model cannot obtain improvements.


In Remark 3, the grassmannn manifold is presented for the cross-view modeling method, how about the two subspaces of source and target domain? What is the difference of the feature distributions before and after domain adaptation, such as t-sne visualization?


It would be suggested to showcase the training process and the computational complexity of the proposed methods individually, such as in the ablation study. Apart from performance, training and runtime are also important for UAVs application.

**Limitations:**

The target domain is limited in only one dataset. There are different UVA datasets public available. But this is not a very large concern, because the experiments are conducted with three different source domains.

---

> ### Author Rebuttal · Authors · 2024-08-05
>
> Dear Reviewer upJj,
>
>
> We greatly appreciate your insightful review and valuable feedback.
> We are very happy you think ***our proposed approach is efficient and achieves significant experimental results***.
> We appreciate your constructive comments and would like to address these points as follows.
>
>
> [Q1] **Comparison with other view transformation**
>
>
> [A1] The method in [1] requires depth for training which is unfaired compared to our approach. Please refer to **[KP1]** for our comparison with other view transformation methods.
>
>
> [Q2] **Results of Advanced Open-vocab Semantic Segmentation**
>
>
> [A2] The training code of clip-as-RNN [2] is not available. Please refer to **[KP2]** for our results using Cat-Seg [3]. Due to the time limitation of the rebuttal period, we leave the investigation of other open-vocab semantic segmentation [4, 5] in our future work.
>
>
>
>
> [Q3] **Difference of UDA settings between CROVIA and EAGLE**
>
>
> [A3] For the unsupervised cross-view adaptation settings, for fair comparison, we follow similar evaluation settings. However, the CROVIA [6] paper only focuses on the cross-view adaptation setting in semantic segmentation. Our work also considers the cross-view adaptation setting on open-vocab segmentation (Tables 5-7).
>
>
> [Q4] **Results of models with and without using cross-view setting**
>
>
> [A4] In Table 2, we have illustrated the performance with and without cross-view adaptation of open-vocab segmentation. In Table 5, we have also reported the performance of DenseCLIP and FreSeg without cross-view (the first row of each group in Table). Please refer to **[KP2]** for additional results of CatSeg.
>
>
> [Q5] **Using estimated camera matrices for the geometric correlation in Eqns (3) and (4)**
>
>
> [A5] Thank you very much for your suggestion. First, we clarify our cross-view modeling approach. From Eqn (3) to (4), it illustrates our approach to cross-view modeling by considering the transformation between two camera views. Then, Eqn (4) illustrates the necessary condition to explicitly model the cross-view geometric correlation under our analysis and assumption mentioned in L182-187. However, we acknowledge that using estimated camera matrices for the geometric correlation in Enq (3) and (4) could be a potential direction for further improvement. Due to the scope of our paper, we leave this investigation as our future work.
>
>
>
>
> [Q6] **Results of other domain adaptation methods on BDD $\to$ UAVID**
>
>
> [A6] In Table 7, we have reported the experimental results of cross-view adaptation setting on prior adaptation methods on BDD $\to$ UAVID, including BiMaL and DAFormer. Due to the time limitation of the rebuttal period, we leave the investigation of other DA methods on BDD $\to$ UAVID as our future work.
>
>
> [Q7] **Reason of DAFormer outperforming Mask2Former.**
>
>
> [A7] Our investigation reveals two reasons that could lead to the lower performance of Mask2Former compared to DAFormer. First, the network backbone of DAFormer is Transformer, while the backbone of Mask2Former in our experiments is ResNet 101. As shown in Table 1, if we use Swin as the backbone of Mask2Former, the results are improved compared to ResNet-101. Second, DAFormer adopts the convolution-based decoder. Meanwhile, Mask2Former adopted a Transformer for the decoder which could require more data for better generalization. Therefore, DAFormer in our experiments performs better than Mask2Former.
>
>
>
>
> [Q8] **Two subspaces of source and target domain and feature distributions before and after adaptation**
>
>
> [A8]  In Remark 3, the Grassmannn manifold is presented and will be used to model the Cross-view Structural Change where the source domain is car-view and the target domain is drone-view. The two subspaces of source and target domain are obtained via the PCA algorithms. The details of subspaces are presented in our Implementation in the appendix (L638-L645). To illustrate the feature distributions, we use the features of the last layer before the classifier in the SYNTHIA $\to$ UAVID experiments. We compare our approach without and with cross-view adaptation. As shown in Figure 3 in the rebuttal pdf, our approach can help to improve the feature representations of classes, and the cluster of each class is more compact, especially in classes of car, tree, and person.
>
>
>
>
> [Q9] **Computational Complexity**
>
>
> [A9] In our proposed approach, the cross-view adaptation loss is only performed during the training. The computational time of our geodesic loss is approximately 0.0195 seconds per iteration. Meanwhile, in the practical deployment, the computational cost of our model relies on the segmentation network. The computational testing cost only relies on the segmentation network (e.g., DeepLab: ~776.2 GFLOPS and DAFormer: ~1447.6 GFLOPS).
>
>
> [Q10] **Target domain is limited in only one dataset**
>
>
> [A10] We have mentioned this limitation in our discussion. Please refer to **[KP3]** for results of an additional target domain.
>
>
>
>
> References
>
>
> [1]  Pan, B., et al. Cross-view semantic segmentation for sensing surroundings. IEEE Robotics and Automation Letters, 2020.
>
>
> [2] Sun, S. et al. Clip as rnn: Segment countless visual concepts without training endeavor. CVPR, 2024.
>
>
> [3] Cho, S., et al. Cat-seg: Cost aggregation for open-vocabulary semantic segmentation. CVPR 2024.
>
>
> [4] Zou, X., et al. Generalized decoding for pixel, image, and language. CVPR, 2023.
>
>
> [5] Wysoczańska, M., et al. Clip-dinoiser: Teaching clip a few dino tricks. arXiv, 2023.
>
> [6] T.D. Truong, et al. CROVIA: Seeing Drone Scenes from Car Perspective via Cross-View Adaptation, arXiv 2024.

---

> > ### Author Response · Authors · 2024-08-13
> > **Rebuttal Follow Up**
> >
> > Dear Reviewer upJj,
> >
> > Thank you so much for your positive rating and insightful feedback!
> >
> > As the reviewer-author discussion is approaching the deadline, we are reaching out to you to ensure that our rebuttal effectively addresses your concerns. If you have any further questions, please let us know. We appreciate your invaluable input.
> >
> > Thank you very much,
> >
> > Authors

---

### Author Rebuttal · Authors · 2024-08-05

## Global Response


We would like to thank all the reviewers for their careful reading and invaluable feedback.
Reviewer JhPV ***accepts*** our paper;
Reviewer upJj and Reviewer 8Etw ***weakly accept***;
and Reviewer 29y1 consider a ***reject*** at this stage.
We appreciate the reviewers encouraged that
***our problem is useful and practical*** (Reviewers 8Etw, JhPV, and 29y1),
***our proposed approach is intuitive and logical, appears to be solid*** (Reviewer 8Etw, JhPV, and 29y1).
and ***our experimental results are solid*** (Reviewers upJj, 8Etw, KhPV, and 29y1).
We have also updated our typos and suggested references in our paper.


On the constructive side, Reviewer 29y1 and Reviewer upJj suggest additional visualization to confirm the effectiveness of the proposed approach. We have included a rebuttal Figure PDF, including a visualization of our qualitative results and feature distributions. In addition, as suggested by Reviewers upJj and 8Etw, to further illustrate the effectiveness of our proposed approach, we conduct additional experiments as follows:


**[KP1] Comparison with Other View Transformation.** To further illustrate our effectiveness compared to other view transformation methods, we compare the results of our approach with another view transformation using DeepLab, i.e., Polar Transformation in [1, 2].  As shown in the table below, our approach outperforms polar transformation.


|  | Road | Building | Car | Tree | Terrain | Person | mIoU |
|---|---|---|---|---|---|---|---|
| BDD $\to$ UAVID |  |  |  |  |  |  |  |
| Polar Transform | 21.1 | 9.6 | 36.4 | 24.1 | 14.6 | 4.6 | 18.4 |
| **EAGLE** | **24.0** | **53.8** | **39.0** | **52.2** | **48.3** | **16.9** | **39.0** |
| SYNTHIA $\to$ UAVID |  |  |  |  |  |  |  |
| Polar Transform | 20.5 | 10.9 | 38.2 | 22.6 | - | 4.3 | 19.3 |
| **EAGLE** | **29.9** | **65.7** | **55.5** | **56.8** | **-** | **18.3** | **45.2** |
| GTA $\to$ UAVID |  |  |  |  |  |  |  |
| Polar Transform | 19.4 | 9.1 | 37.8 | 20.7 | 15.6 | 2.5 | 17.5 |
| **EAGLE** | **20.5** | **53.0** | **37.6** | **50.7** | **45.3** | **13.0** | **36.7** |


**[KP2] Results of Advanced Open-vocab Semantic Segmentation.** We reproduce the results of Cat-Seg [3] as shown in the following table. Overall, our cross-view adaptation can significantly improve the performance of open-vocab semantic segmentation models.


|  | Road | Building | Car | Tree | Person | mIoU |
|---|---|---|---|---|---|---|
| CatSeg | 19.9 | 27.3 | 22.8 | 33.1 | 10.7 | 22.8 |
| CatSeg+CrossView | 37.2 | 73.0 | 61.8 | 61.0 | 20.1 | 50.6 |
| EAGLE | 36.8 | 75.5 | 61.3 | 60.8 | 21.2 | 51.1 |
| EAGLE+ViewCondition | **38.4** | **76.1** | **62.8** | **62.1** | **21.8** | **52.2** |


**[KP3] Results of Additional Target Dataset**. To further illustrate the effectiveness of our approach, we conduct an additional cross-view adaptation using the UDD dataset [4], i.e., SYNTHIA $\to$ UDD, with 4 classes of 'Tree', 'Building', 'Road', and 'Vehicle'. We adopt the DeepLab segmentation network in our experiments. As shown in our experiments, our approach outperforms the prior adaptation method and closes the gap with supervised learning.




|  | Tree | Building | Road | Vehicle | mIoU |
|---|---|---|---|---|---|
| No Adapt | 19.66 | 13.50 | 9.66 | 6.55 | 12.34 |
| BiMaL | 35.61 | 29.71 | 21.17 | 19.05 | 26.38 |
| **EAGLE** | **48.27** | **44.07** | **41.39** | **38.60** | **43.08** |
| Upper Bound | 70.37 | 77.79 | 78.42 | 68.56 | 73.79 |




In addition, we have provided comprehensive responses to queries raised by the reviewers. We hope our explanations can effectively address the reviewers' concerns. Once again, we want to thank all the reviewers for their valuable and insightful feedback. We sincerely hope that our efforts will result in a favorable reconsideration of the scores by the reviewers.


References


[1] Y. Shi, et al. Spatial-aware feature aggregation for image based cross-view geo-localization. NeurIPS, 2019.


[2] Y. Shi, et al. Where am I looking at? joint location and orientation estimation by cross-view matching. CVPR, 2020.


[3]  Cho, S., et al. Cat-seg: Cost aggregation for open-vocabulary semantic segmentation. CVPR 2024.


[4] Y Chen, et al. Large-scale structure from motion with semantic constraints of aerial images. PRCV, 2018.

---

### Decision · Program_Chairs · 2024-09-25

**Decision:**

Accept (poster)

**Comment:**

Two of the reviewers recommended weak accept, and the others accept and reject. While 29y1 recommended rejection, there comments during the discussion after reading the rebuttal indicated that their only remaining concern was regarding the lack of qualitative comparisons to more recent methods.

Given the above, this AC agrees with the reviewers and recommends that the paper is accepted. The authors are strongly encouraged to take the reviewer recommendations and suggestions into account when revising the paper for the camera ready version. Specifically, they should focus on the following:
* Update the text to fix the cases where the text/descriptions/notation were not clear to the reviewers.
* The paper is very cramped, use the extra page in the camera ready version to space it out more so that it is more readable.
* Add the comparisons to other view transformation methods provided to upJj.
* Add the additional qualitative comparisons to newer methods (i.e. not just the older methods displayed in the rebuttal) requested by 29y1 and 8Etw. Fig. 3 from the rebuttal PDF is fine for the appendix, but not necessary to add it to the main paper.
* Update the related work text so that the difference to Crovia by Truong et al. is more clearly defined.

Minor comments:
* Fig. 1 caption “could not perform well” -> “do not perform well”
* Fig. 2 It is not clear what method is being shown here. Improve the caption.